# Modular fluorescence complementation sensors for live cell detection of epigenetic signals at endogenous genomic sites

Cristiana Lungu[1], Sabine Pinter[1], Julian Broche[1], Philipp Rathert[1] & Albert Jeltsch [1]

Investigation of the fundamental role of epigenetic processes requires methods for the locus-specific detection of epigenetic modifications in living cells. Here, we address this urgent demand by developing four modular fluorescence complementation-based epigenetic biosensors for live-cell microscopy applications. These tools combine engineered DNA-binding proteins with domains recognizing defined epigenetic marks, both fused to non-fluorescent fragments of a fluorescent protein. The presence of the epigenetic mark at the target DNA sequence leads to the reconstitution of a functional fluorophore. With this approach, we could for the first time directly detect DNA methylation and histone 3 lysine 9 trimethylation at endogenous genomic sites in live cells and follow dynamic changes in these marks upon drug treatment, induction of epigenetic enzymes and during the cell cycle. We anticipate that this versatile technology will improve our understanding of how specific epigenetic signatures are set, erased and maintained during embryonic development or disease onset.

---

[1] Department of Biochemistry, Institute of Biochemistry and Technical Biochemistry, Stuttgart University, Allmandring 31, 70569 Stuttgart, Germany. Correspondence and requests for materials should be addressed to A.J. (email: albert.jeltsch@ibc.uni-stuttgart.de)

Epigenetic modifications such as DNA methylation and post-translational modifications of histone proteins are critical contributors to the reprogramming and maintenance of cellular states during development or disease. Although they do not alter the primary DNA sequence, epigenetic marks regulate chromatin functions including gene expression, in a dynamic and genomic context-specific manner[1–4]. Centromeric mouse major satellites and human α-satellites are archetypical spots of constitutive heterochromatin where DNA cytosine-C5 methylation (5mC) and tri-methylation of lysine 9 on histone H3 (H3K9me3) are enriched[5]. In diseases such as cancer repetitive sequences including heterochromatic DNA repeats, dispersed retrotransposons, and endogenous retroviral elements, become frequently hypomethylated, while CpG islands of tumor suppressor genes often gain DNA methylation[6, 7]. Hence, a deeper understanding of the molecular functions and biological roles of epigenetic marks requires the sequence-specific investigation of these signals. Furthermore, since the epigenetic landscape is highly dynamic during cellular differentiation and pathological development, a meaningful interpretation of epigenetic signaling cascades can only be obtained by combining the static information on the locus-specific status of epigenetic marks with a real-time readout of their changes.

A comprehensive understanding of epigenetic signaling cascades is hindered by the lack of methods that enable a dynamic and targeted readout of epigenetic modifications in living cells at the level of endogenous loci. Affinity-based enrichment methods are frequently employed to map the genome-wide distributions of 5mC and histone modifications[8, 9] but these procedures require cell lysis, thereby providing only a snapshot of the dynamic epigenetic landscape and obstructing information on cellular physiology. In histological sections, locus specific readout of histone marks has been addressed in a proximity ligation assay by combining antibody detection of the epigenetic mark with fluorescence in situ hybridization (FISH) for locus resolution[10].

Alternatively, 5mC readout was achieved by coupling FISH with 5mC-specific crosslinking of the probe with osmium tetroxide[11]. Nevertheless, both of these methods provide only a static snapshot of the epigenetic state and require harsh chemical treatment, which makes them incompatible with live-cell applications. To assess the status of epigenetic marks in live cells, fluorophore-coupled affinity probes for real-time tracking of epigenetic modifications were used[12–15]. However, all these microscopic tools are currently restricted to imaging only global changes of the targeted epigenetic modification and have no DNA sequence resolution.

To overcome these methodological limitations, we engineered an epigenetic detection method for dynamic and direct readout of locus-specific epigenetic signals in live mammalian cells using modular fluorescence complementation-based BiAD (Bimolecular Anchor Detector) sensors consisting of anchor modules for programmable sequence-specific DNA binding and detector domains for chromatin mark recognition. Readout of the signal was based on bimolecular fluorescence complementation (BiFC)[16].

With this approach, we could for the first time to the best of our knowledge, directly detect locus-specific changes of pericentromeric 5mC and H3K9me3 levels in living cells. The BiAD sensors are specific, modular and robust, and can be used in various combinations and different cell types. We anticipate that these versatile tools will set the basis for a better understanding of epigenetic signaling cascades that occur during cellular development, re-programming, response to drugs or pathological changes.

## Results

**Sensor design.** To achieve a specific readout of target epigenetic modifications with genomic locus resolution, we designed a set of modular BiFC-based sensors (Fig. 1). These consist of an anchor module, for DNA sequence-specific recognition, and a detector

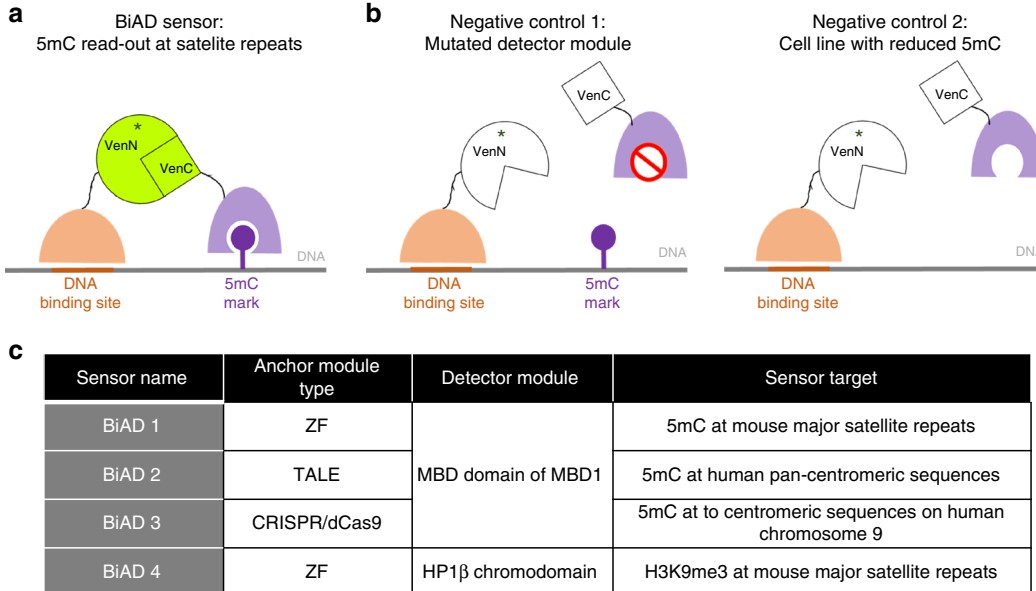

| Sensor name | Anchor module type | Detector module | Sensor target |
|---|---|---|---|
| BiAD 1 | ZF | | 5mC at mouse major satellite repeats |
| BiAD 2 | TALE | MBD domain of MBD1 | 5mC at human pan-centromeric sequences |
| BiAD 3 | CRISPR/dCas9 | | 5mC at to centromeric sequences on human chromosome 9 |
| BiAD 4 | ZF | HP1β chromodomain | H3K9me3 at mouse major satellite repeats |

**Fig. 1** Design concept and experimental validation strategy of the BiAD sensors. **a** An anchor domain (shown in *orange*) is used to recognize a specific genomic locus (shown as *orange line*) and a detector domain (shown in *purple*) is employed for the recognition of a target epigenetic modification (shown as *lolly pop*). Both proteins are fused to the non-fluorescent VenN and VenC parts of mVenus. The position of the chromophore is schematically indicated with a *star* within the VenN part. When the targeted DNA sequence is methylated, the two domains will bind in close spatial proximity, leading to the reconstitution of a functional mVenus fluorophore. This can be visualized by fluorescence microscopy. **b** If the detector module is deactivated by a mutation in the 5mC-binding pocket, or the sensor is expressed in cells with reduced 5mC levels, no fluorescence complementation signal is observed. **c** Overview of the sensors designed during the course of this study

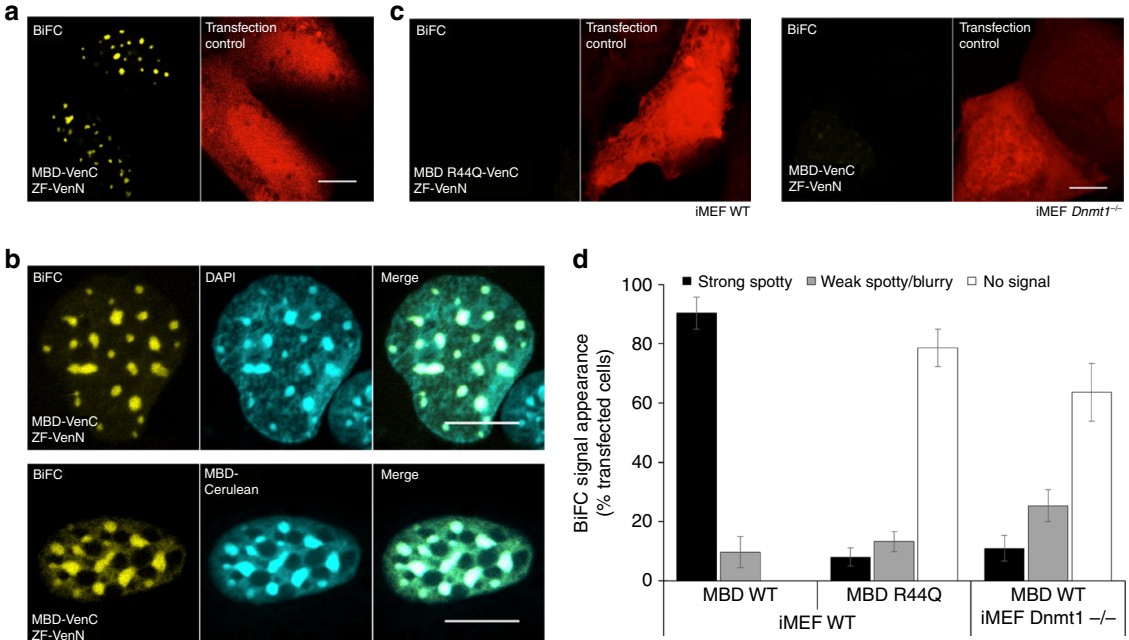

**Fig. 2** Development and validation of the BiAD sensor 1. **a** Representative fluorescence microscopy image of the BiFC signal (*yellow channel*) generated upon transfection of the BiAD sensor 1 in iMEF cells. A plasmid encoding NLS-mRuby2 was used to identify transfected cells (*red channel*). **b** Representative fluorescence microscopy images documenting co-localization of the BiFC signal with DAPI stained major satellite DNA (*upper panel*) and 5mC marks detected by co-transfection of the sensor modules with MBD-Cerulean (*lower panel*). **c** Representative fluorescence microscopy images documenting the 5mC specificity of the BiAD sensor. The BiFC signal was lost with the MBD R44Q 5mC-binding pocket mutant (*left*) and in cells with globally reduced DNA methylation levels (*right*). A plasmid encoding NLS-mRuby2 was used to identify transfected cells (*red channel*). The transfection, imaging and display settings of the images shown in panels **a**, **c** are identical. **d** Quantification of the experiments representatively shown in panels **a**–**c**. The *error bars* represent the s.e.m. for two biological repeats (for details *cf.* Methods and Supplementary Tables 1, 6, and 7). All cells were fixed at 48 h after transfection. *Scale bar* for all images is 10 μm

module, which specifically binds to defined chromatin modifications. Previously validated Zinc-finger, TAL effector and CRISPR-dCas9 systems were employed as anchor modules with high-sequence specificity[17–20] and the MBD of MBD1[21] and chromodomain of HP1β[22] were used as detector modules for 5mC and H3K9me3. Both the anchor and detector modules were fused to the non-fluorescent N- and C-terminal fragments of monomeric Venus[23, 24]. If the target locus carries the epigenetic modification of interest, binding of the anchor and detector modules in close spatial proximity leads to the reconstitution of a functional Venus fluorophore, which emits a stable fluorescent signal that can be microscopically tracked (Fig. 1a). The dependence of the different biosensors generated here on their target chromatin modifications was tested by employing binding pocket mutations in the detector module, as well as cell lines with globally reduced levels of the investigated epigenetic marks. In both control settings, the stable docking of the detector module on chromatin was expected to be impaired, and efficient fluorophore reconstitution should be strongly reduced (Fig. 1b). An overview of the sensors developed in this work is provided in Fig. 1c.

**BiAD 1 specifically reads 5mC at major satellites**. To establish the BiAD sensor system, we initially focused on the detection of 5mC at mouse major satellite repeats[25]. These pericentromeric loci are archetypical sites of 5mC enrichment and form highly abundant tandem repeat arrays that localize into distinguishable 4′,6-diamidino-2-phenylindole (DAPI)-dense foci (Supplementary Fig. 1a). We adapted a zinc finger (ZF) protein that was previously used to visualize the 5′-GGCGAGGAA-3′

motif within mouse major satellite repeat sequences[17, 26, 27]. Upon transfection into NIH3T3 cells, the ZF-Venus fusion showed complete overlap with DAPI staining (Supplementary Fig. 1a). This localization pattern is in agreement with previous reports and confirms the sequence specificity of the anchor module[17, 27]. To detect the 5mC mark, the fluorophore-fused MBD of MBD1 was used. This protein was previously employed for specific tracing of methylated DNA in cultured cells and for generating a 5mC mouse reporter model[12, 28]. It has been documented to show a high 5mC specificity both in vitro and in vivo, combined with minimum cellular toxicity[12, 29–32]. Moreover, biochemical work indicated that the protein is able to bind double stranded methylated DNA in different sequence contexts[33]. As expected, in pilot experiments with Venus fused MBD we observed an accumulation of the fusion protein at DAPI-dense heterochromatic foci in transfected mouse fibroblats (Supplementary Fig. 1b). Furthermore, in co-transfection experiments, a clear co-localization between the anchor and the detector module (each fused to a full fluorophore) was observed (Supplementary Fig. 1c). This indicates that compacted heterochromatin can be accessed by both proteins and there is no evident competition of the two modules for binding sites.

For BiAD sensor 1, the anchor and detector modules were fused with non-fluorescent complementary fragments of Venus to set up a BiFC system where detectable fluorescence can occur if both parts approach each other at intermolecular distances as low as 10 nm[34]. This dramatically improves the resolution with which the epigenetic mark can be detected. After ensuring that fusion to the split Venus fragments did not negatively affect the localization, and thus the specificity, of the anchor and detector modules (Supplementary Fig. 2a, b), we co-transfected both

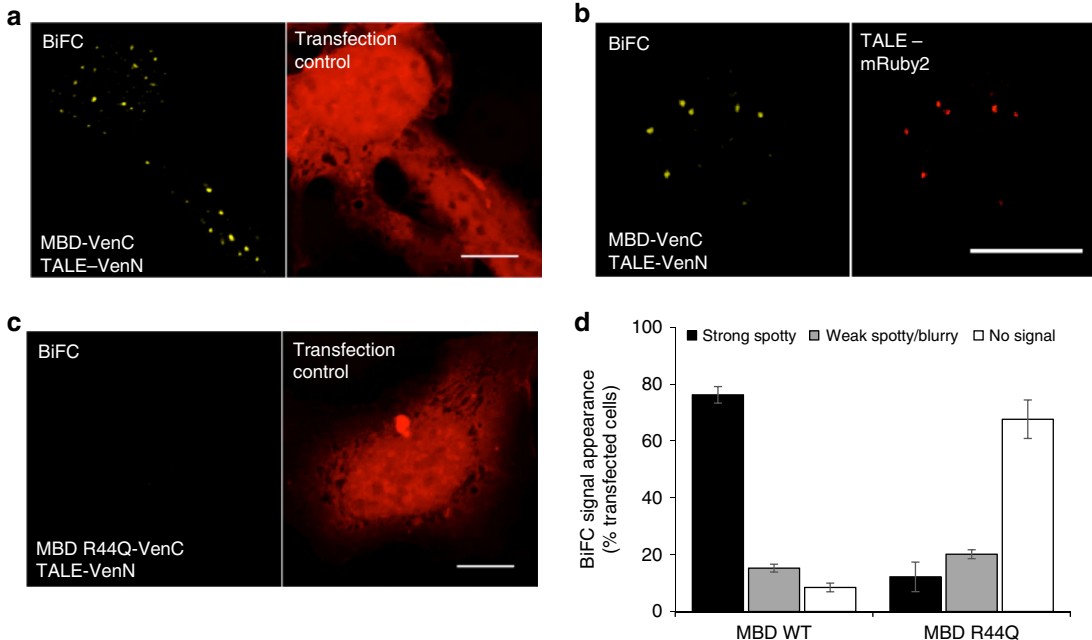

**Fig. 3** Development and validation of the BiAD sensor 2. **a** Representative fluorescence microscopy image showing that BiAD sensor 2 gives rise to distinct BiFC signals in HEK293 cells. NLS-mRuby2 was used as transfection control. **b** Co-localization of the BiFC signal with the TALE protein fused to mRuby2 documents the high-DNA specificity of the BiAD sensor. **c** Loss of the BiFC signal when the R44Q MBD variant was used as a detector module confirms the 5mC specificity of the BiAD sensor. **d** Quantification of the experiments representatively shown in panels **a**–**c**. The *error bars* represent the s.e.m. for two biological repeats (for details cf. Methods and Supplementary Tables 2, 6, and 7). All cells were fixed at 48 h after transfection. *Scale bar* for all images is 10 μm

domains in mouse cells. We observed strong BiFC with a high signal to background ratio in both live and fixed cells indicating that the sensor has an excellent reconstitution yield (Fig. 2a and Supplementary Figs. 2, 3, and 4). Overall, we robustly detected strong BiFC signals in around 90% of the transfected cells (Fig. 2d). Since we did not observe a negative effect of cellular fixation on the BiFC signal (Fig. 2a vs. Supplementary Fig. 3a), we moved on with analyzing the performance of the BiAD sensor in fixed cells.

Co-localization of the BiFC signal with DAPI foci validated the DNA sequence specificity of the sensor (Fig. 2b, *upper panel*). To prove the 5mC specificity of the tool, we have co-transfected the BiAD detector modules with MBD-Cerulean as a marker for 5mC. We observed that the BiFC signal was formed only at sites also bound by MBD-Cerulean (Fig. 2b, *lower panel*). Altogether, these results show that the sensor can be used for direct imagining of 5mC on the targeted DNA sequence in live as well as fixed cells.

To further confirm the 5mC specificity of the complementation signal, we have next inactivated the methyl-binding pocket in the MBD detector module by exchanging the conserved R44 residue in the 5mC-binding hydrophobic patch to Q, which prevents 5mC binding[35]. In contrast to the WT construct, the R44Q variant showed a predominantly diffuse nuclear localization with no enrichment at DAPI foci, in line with its loss of 5mC binding (Supplementary Fig. 5a). This pattern was maintained upon fusion of the detector module with the Venus C-terminal fragment (Supplementary Fig. 5b, c). When the MBD R44Q variant detector domain was used in the BiFC assay, we observed a dramatic reduction in the intensity of the reconstituted fluorescence signal in live as well as fixed mouse fibroblasts (Fig. 2c, d and Supplementary Figs. 3b, 6). Importantly, this was not due to the altered stability of the MBD R44Q variant, as both the wild-type and the mutant displayed comparable expression levels (Supplementary Fig. 5c). These results indicate that a fully

functional MBD domain is essential for productive fluorescence reconstitution in the context of the BiAD sensor. Furthermore, they clearly document that the formation of BiFC signals depends on the stable and specific docking of both BiAD modules to chromatin and it is not caused by random associations between the diffusing detector modules and the chromatin bound anchor modules.

As a last validation for the specificity of the BiAD sensor we employed *Dnmt1*$^{-/-}$ iMEF cells, which have a strongly reduced DNA methylation at the targeted genomic sites[26]. Indeed, transfection of MBD-Venus into this cell line gave rise to a predominantly diffuse localization pattern, although pericentromeric heterochromatin foci where intact as indicated by DAPI staining (Supplementary Fig. 7a). This change in localization was a direct effect of the 5mC reduction and not of chromatin reorganization (Supplementary Fig. 7b). In line with this, transfection of the BiAD sensor into *Dnmt1*$^{-/-}$ iMEFs led to low levels of fluorescence reconstitution, while normal BiFC levels were observed in Suv39DKO cells. The drop of the BiFC signal observed in the *Dnmt1*$^{-/-}$ iMEFs was similar to what was obtained when the 5mC-binding deficient MBD R44Q variant was used as a detector module (Fig. 2c, d and Supplementary Figs. 3b, 8, 9). Altogether, these results indicate that the BiAD sensor 1 can be used to directly visualize the status of 5mC at repetitive sequences in murine cells.

**BiAD 2 specifically reads 5mC at human α-satellites.** To study the chromatin of human cells we developed BiAD sensor 2 detecting 5mC at centromeric alpha-satellite sequences[36]. By taking advantage of the modularity of the BiAD sensor, we could integrate the validated 5mC detector module into the human sensor (Supplementary Fig. 10). As anchor module, we used a TALE protein already demonstrated to specifically recognize the pan-centromere target sequence 5′-TAGACA-GAAGCATTCTCAGA-3′[18]. The localization of the split-

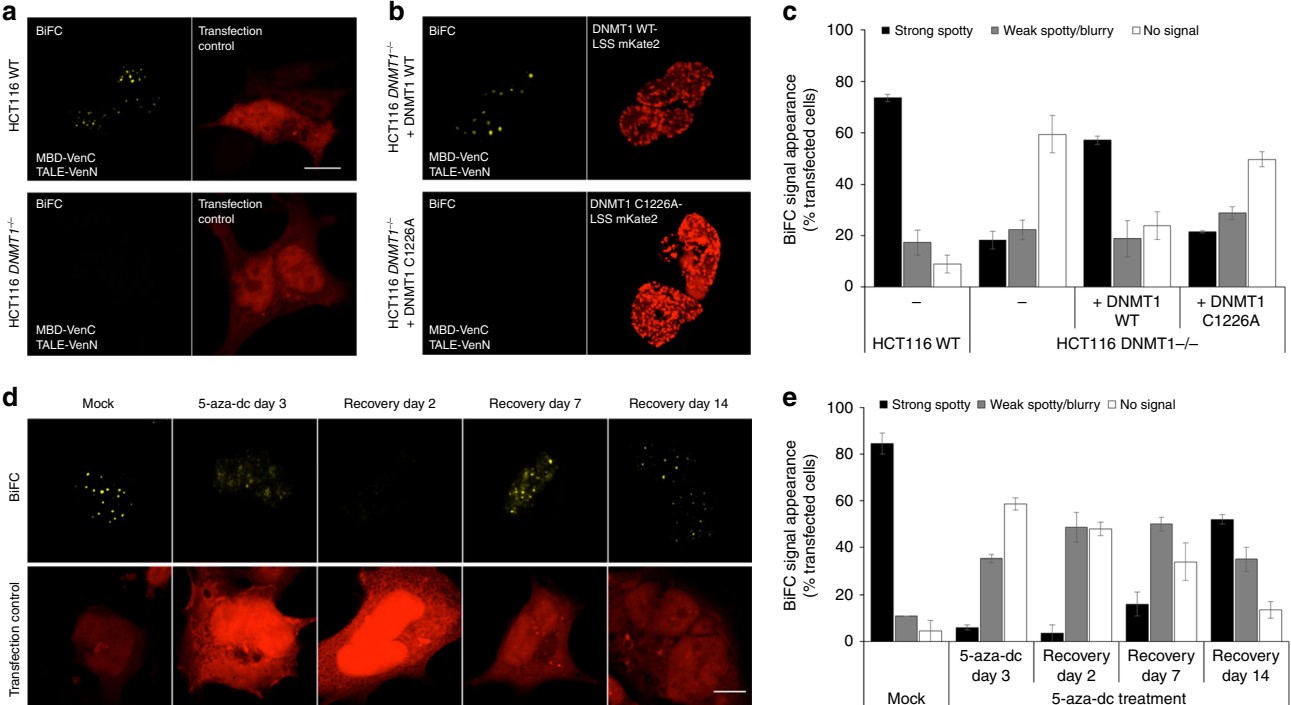

**Fig. 4** Visualization of DNA methylation changes using BiAD sensor 2. **a** Representative fluorescence microscopy images documenting that strong BiFC signals are formed upon transfection of the BiAD sensor 2 into HCT116 WT cells (*top*). In contrast, very low fluorescence reconstitution was observed in DNMT1 hypomorphic HTC116 cells that have globally reduced methylation levels (*bottom*). A plasmid encoding for NLS-mRuby2 was used to identify transfected cells. **b** The BiFC signals were rescued in the DNMT1 hypomorphic HTC116 cells by the exogenous expression of active DNMT1 (*top*) but not of catalytically inactive DNMT1 (*bottom*). **c** Quantification of the experiments representatively shown in panels **a**, **b**. **d** Time course of DNA demethylation by treatment of HEK293 cells with 5-aza-dC and recovery. In each sample, the BiAD sensor was transfected 2 days before the cells were fixed for imaging. **e** Quantification of the experiments representatively shown in panel **d**. *Error bars* in all images represent the s.e.m. for two biological replicates (for details *cf.* Supplementary Tables 2, 6, and 7). *Scale bar* for all images is 10 µm. The imaging and display setting of the BiFC images shown in panels **a**, **b**, and **d** are identical. In panel **d** the contrast of the mRuby2 channel was increased to enhance the visibility of transfected cells. This was done with the same settings for all time points

fluorophore fused TALE protein was identical to the pattern obtained for TALE-Venus (Supplementary Fig. 11b). This indicates that the split-fluorophore does not change the DNA sequence specificity of the anchor device and that the extensive validation of this DNA-binding protein[18] can be extrapolated to the split Venus fusion used in BiAD sensor 2 (Supplementary Fig. 11a). No BiFC signal was detectable when the split fluorophore-fused anchor module was transfected alone in HEK293 cells (Supplementary Fig. 11c). Unlike in mouse cells, the pericentromeric heterochromatin of human cells is not organized in DAPI-dense structures[37]. Accordingly, the 5mC detector module displayed a fine granular pattern in HEK293 cells. This was independent of whether the MBD was expressed as a full-fluorophore or a split-Venus fusion (Supplementary Fig. 10a vs. b) and it is comparable with the patterns previously observed after 5mC antibody staining in this cell line[38]. The localization of the MBD R44Q variant was diffuse with occasional enrichment in nucleoli (Supplementary Fig. 10).

Since the efficiency of the BiFC signal was shown to be sensitive to local steric hindrances[16], several variants of the sensor were designed with the VenN and VenC fragments fused at either the N- or C- terminus of the MBD and TALE and with a longer linker separating the TALE from the fluorophore. Transfection of these variants in HEK293 cells resulted in BiFC signals with strikingly different yields (Supplementary Fig. 12). In general, a fluorescent signal was only obtained when the detector module was fused with the C- terminus of Venus (Supplementary Fig. 12c vs. d). The lack of BiFC signal formation of the VenN-MBD fusions could not be attributed to mis-folding or delocalization, as immunofluorescence

images showed a granular nuclear localization of all split-fluorophore fusions similar as with the full fluorophore-fused domain (Supplementary Figs. 12a, 10). Based on its high signal-to-noise ratio, the TALE-VenN and MBD-VenC pair was selected for all further experiments (Fig. 3a). Remarkably, this optimized BiAD sensor 2 resulted in strong Venus reconstitution in circa 75% of the transfected cells (Fig. 3d). The punctuate and virtually background-free BiFC signal observed in HEK293 cells upon co-transfection of the anchor/detector modules highlights the advantage of the BiAD approach over co-localization methods such as immunofluorescence-FISH (Supplementary Fig. 13a vs. b).

We next co-transfected the BiAD sensor together with TALE-mRuby2 and observed a strong correlation of the BiFC and red fluorescence signals (Fig. 3b) validating the DNA sequence specificity of the BiAD sensor. To evaluate the 5mC-dependence of the reconstituted fluorescence signal, we used the MBD R44Q-binding pocket mutant and observed a fivefold reduction in the number of transfected cells that showed a BiFC signal (Fig. 3c, d and Supplementary Fig. 14). These results demonstrate that the newly developed BiAD sensor 2 is specific for the detection of the 5mC mark at alpha satellites in human cells.

**BiAD 2 detects locus-specific changes in 5mC levels**. To investigate dynamic changes in 5mC level with the BiAD 2 sensor the HCT116 DNMT1 hypomorphic cells were used, which contain a truncated DNMT1 with reduced activity and were shown to have a 20% decrease in the global levels of DNA methylation[39–41]. Target bisulfite amplicon-based next generation

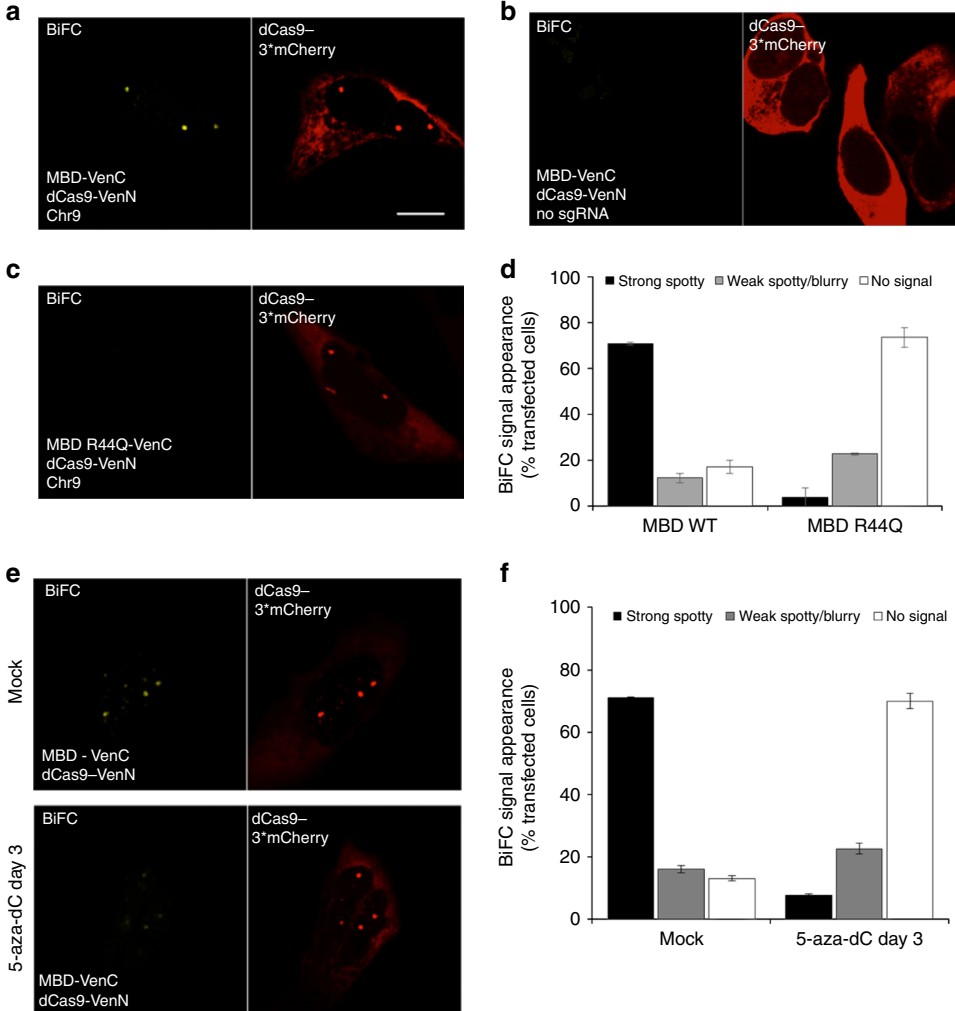

**Fig. 5** Development and validation of the BiAD sensor 3. **a** Representative fluorescence microscopy image showing strong and sequence-specific BiFC signal obtained with BiAD sensor 3. The locus-specificity of the BiFC signal was validated by co-transfecting the BiAD modules with a red fluorophore-tagged dCas9. **b** Loss of BIFC signal in the absence of the cognate sgRNA. **c** Loss of BiFC signal with the R44Q MBD mutant. **d** Quantification of the experiments representatively shown in panels **a–c**. **e** Representative fluorescence microscopy images documenting the changes in the intensity of the BiFC signal after a three-day 5-aza-dC treatment of sensor-transfected HEK293 cells. **f** Quantification of the experiments representatively shown in panel **e**. *Error bars* in all images represent the s.e.m. for two biological replicates (for details *cf.* Supplementary Tables 3, 6, and 7). All cells were fixed at 48 h after transfection. *Scale bar* for all images is 10 μm. The imaging and display settings of the images shown in panels **a–c** and within **e** are identical

sequencing revealed a 40% reduction in the levels of DNA methylation near the TALE-binding site in the HCT116 DNMT1 hypomorphic cell line, clearly indicating that this cell line is a suitable model system for testing the sensitivity of the BiAD sensor 2 (Supplementary Fig. 15). Similar to HEK293 cells, strong and specific BiFC signals were observed in HCT116 WT cells carrying an intact DNMT1 protein (Fig. 4a, c). This was in stark contrast to the fourfold reduction in the number of cells showing a strong and spotty BiFC signal in the hypomorphic cell line (Fig. 4a, c). This result confirms the direct dependence of the BiAD sensor on the presence of the 5mC mark and its response to changes within a physiological range.

Next, we co-transfected catalytically active DNMT1 together with the BiAD 2 modules into the DNMT1 hypomorphic cells (Fig. 4b). Expression of all proteins and methylation recovery was allowed to proceed for 2 days before the cells were imaged. Under these conditions, a threefold increase in the number of cells showing a strong spotty BiFC signal was observed (Fig. 4c). Importantly, no signal increase was observed when the catalytically inactive DNMT1 C1226A variant was used (Fig. 4b, c).

We next aimed to monitor changes of 5mC levels at the target loci upon drug treatment. To this end, HEK293 cells were treated with 5-aza-2′-deoxycytidine (5-aza-dC), an established inhibitor of DNA methylation (Supplementary Fig. 16a)[42]. No obvious localization differences of the TALE-Venus anchor module were observed in mock or 3-day 5-aza-dC-treated cells (Supplementary Fig. 16b). As expected, the localization of the MBD WT and R44Q variant did not detectably change in human cells upon 5-aza-dC treatment (Supplementary Fig. 16c). We next co-transfected the BiAD modules in mock and drug-treated HEK293 cells and traced the recovery of DNA methylation for 2 weeks after removing the inhibitor by performing serial transfections with the detector modules (Fig. 4d, e). A strong decrease in the BiFC signal was observed after 3 days 5-aza-dC treatment, which was still detectable 2 days after drug removal (Fig. 4e). However, 7 days after drug removal increasing levels of methylation were visualized with the BiAD sensor followed by a further increase after 2 weeks of recovery (Fig. 4e). A comparable re-methylation kinetics was previously observed for Alu-repeats[43]. Further control experiments showed that the signal changes observed

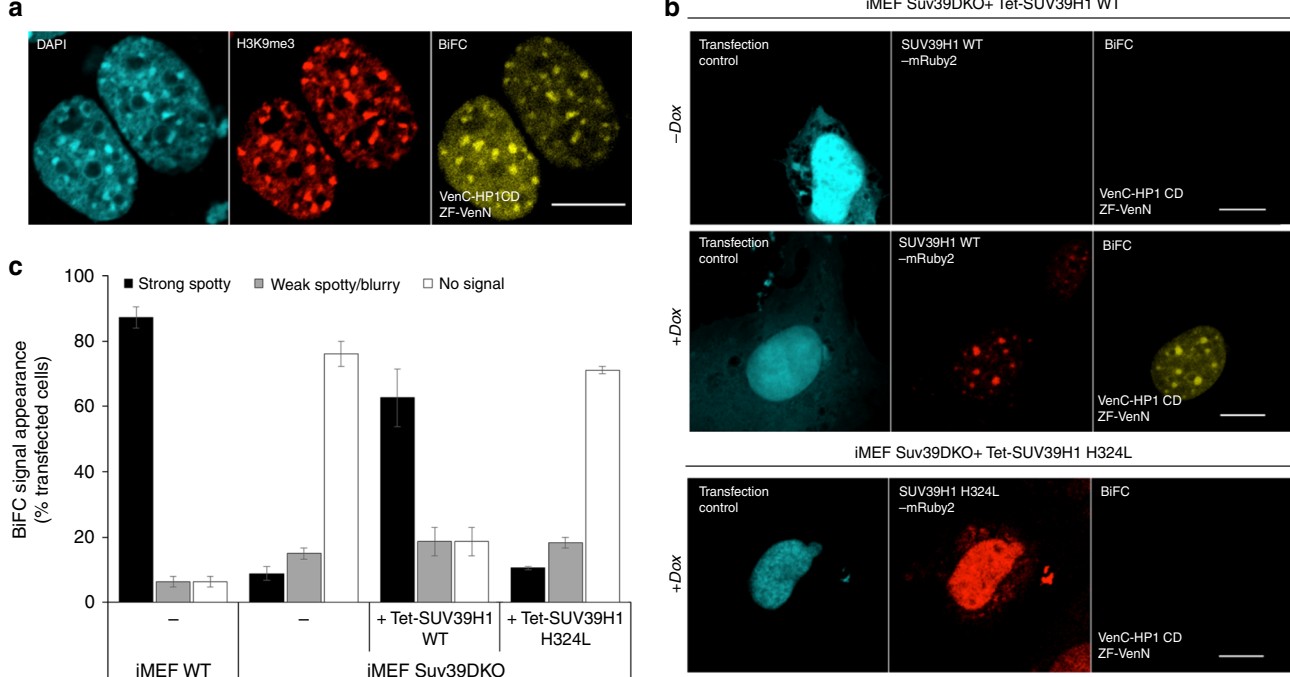

**Fig. 6** Readout of changes of H3K9me3 at mouse major satellite repeat sequences using BiAD sensor 4. **a** Representative fluorescence microscopy images demonstrating the co-localization of BiFC signal, DAPI staining, and H3K9me3 antibody staining. **b** Application of the BiAD sensor to detect changes in H3K9me3 levels after dox induced expression of SUV39H1 in Suv39DKO iMEF cells. A BiFC signal could only be observed when the sensor was transfected in cells expressing SUV39H1 (*middle panel*). No BiFC signal was detected in cells expressing the catalytically inactive H324L SUV39H1 mutant (*bottom panel*), **c** Quantification of the experiments representatively shown in panels **a**, **b**. Error bars represent s.e.m. for two biological replicates (for details *cf.* Supplementary Tables 4 and 6). *Scale bar* for all images is 10 μm. All cells were fixed 48 h after transfection and the imaging and display settings of the images shown **b** are identical

with the BiAD sensor were independent of changes in chromatin organization caused by the 5-aza-dC treatment (Supplementary Fig. 17). This observation indicates that the fluorescent signal produced by the BiAD sensor 2 arises through the binding of the BiAD modules next to each other on chromatin and it is not caused by three-dimensional (3D) reconstitution events, which would be strongly affected by the global chromatin structure.

**BiAD 3 detects changes in 5mC levels at Chr9 α-satellites**. To further increase the locus specific resolution of our detection system we used the programmable Sp-dCas9 protein, together with a single guide (sg) RNA previously employed for specific targeting of a pericentromeric sequence located only on human chromosome 9[19]. Upon transfecting the dCas9-Venus fusion together with the sgRNA in HEK293 cells, we consistently observed 2–4 bright foci in each cell nucleus (Supplementary Fig. 18), which is in line with the polyploidy of these cells[44]. Next, the dCas9 protein was fused to VenN and used as anchor domain, together with the MBD-VenC detector module (BiAD sensor 3). Co-transfecting these BiAD modules gave rise to bright BiFC signals that fully co-localized with red fluorescent foci observed after triple cotransfection with dCas9 fused to three copies of mCherry (dCas9-3*mCherry) (Fig. 5a). The BiFC signal was completely dependent on the presence of the sgRNA (Fig. 5b). The 5mC-specificity of the novel sensor was validated using the MBD R44Q mutant detector module. With the MBD mutant we observed a 15-fold decrease in the number of cells forming strong and spotty fluorescent signals (Fig. 5c, d and Supplementary Fig. 19). To evaluate the dynamic range of the novel BiAD sensor 3, we used 5-aza-dC to induce global DNA demethylation in the HEK293 cells. After 3 days of drug treatment, we observed a ten-fold decrease in the number of strong BiFC-positive

cells (Fig. 5e, f). Importantly, this decrease in fluorescence reconstitution was not due to mis-targeting of the dCas9 upon drug treatment, as the localization of the co-transfected dCas9-3*mCherry marker did not change. These data clearly demonstrate that dCas9 efficiently functions as BiAD anchor module.

**BiAD 4 detects changes in H3K9me3 levels at major satellites**. To develop a BiAD system to monitor histone tail marks at specific genomic loci, we focused on the detection of hetero-chromatic H3K9me3, which is introduced by the SUV39H1 and SUV39H2 protein lysine methyltransferases[45] and abundantly decorates facultative and constitutive heterochromatin[46]. For DNA sequence recognition in the BiAD sensor 4, we used the ZF targeting mouse major satellite repeats. To detect H3K9me3, the chromo domain of HP1β (HP1CD) was selected, which retains the high H3K9me3-binding affinity of the full length protein[47] while lacking the SUV39H1-interacting chromoshadow domain[48]. The H3K9me3 specificity of this detector was confirmed by H3K9me3 antibody stain in WT (Supplementary Fig. 20a) and Suv39DKO iMEF cells (Supplementary Fig. 21a, b), which were in agreement with previous studies using the full length HP1β protein[49]. Consistent with the unaltered geometry of DAPI-dense chromocenters in the Suv39DKO cells reported previously[50], the anchor module maintained its spotty localiza-tion (Supplementary Fig. 21b). Adapting the chromo domain into the BiAD sensor by fusion with the C- terminus of Venus did not influence protein localization (Supplementary Fig. 20b).

The new detector module was combined with the ZF anchor for locus-specific H3K9me3 detection. Several versions of the two modules were generated, where the position of the split Venus fragment was shuffled relative to the two domains. While BiFC

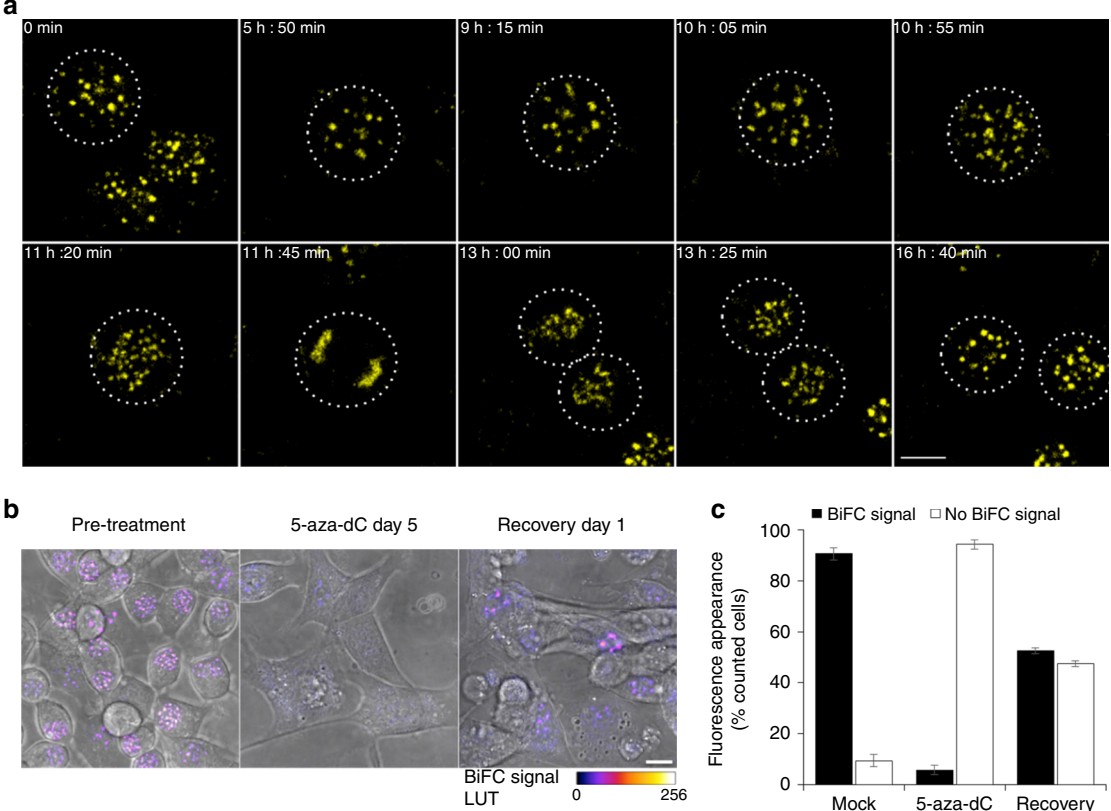

**Fig. 7** Real-time visualization of DNA methylation levels on major satellite repeats with BiAD sensor 1. **a** Live-cell imaging of cells stably expressing the BiAD sensor 1 modules. The time points at which images were taken are annotated. The panels are centered on the cell that undergoes mitosis during the imaging interval. To facilitate visualization, this cell was marked with a *dotted circle*. The imaging and data display conditions are identical between the shown time points. **b** Representative phase contrast-fluorescence channel overlay images during the 5-aza-dC DNA demethylation treatment of the cell line stably expressing BiAD sensor 1. To facilitate visualization, the BiFC signal was pseudo colored. The imaging and data display conditions are identical between the three panels of the composite image. **c** Quantification of the experiments representatively shown in panel. *Error bars* represent s.e.m. for two biological replicates (for details *cf.* Supplementary Tables 6 and 7). *Scale bar* for all images is 10 μm

signals were visible for all tested combinations, the ZF-VenN and VenC-HP1CD pair displayed the highest signal-to-noise ratio and was, therefore, selected for further applications (Supplementary Fig. 22). The modification and DNA sequence-specificity of the novel BiAD sensor was confirmed by H3K9me3 and DAPI co-staining (Fig. 6a). Remarkably, despite the higher mobility of histone tails, approx. 85% of the transfected cells displayed a strong BiFC signal (Fig. 6c). This was comparable to what was obtained for the 5mC readout at these sites and documents the general applicability of BiAD sensors. To further validate the H3K9me3 specificity of this sensor, we have exchanged the conserved W42 residue within the HP1 chromodomain to A to generate a detector domain that is deficient in H3K9me3 binding[51]. In line with this, transfection of either Venus or VenC-tagged HP1CD W42 domain into murine cells revealed that the W42A variant no longer localizes to mouse chromocenters (Supplementary Fig. 23a, b). Subsequent BiFC assays with BiAD sensor 4 using HP1CD W42A as a detector module revealed a dramatic drop in the percentage of cells that showed a strong, spotty BiFC signal (88% vs. 2%) (Supplementary Fig. 23c, d). Altogether, this series of validation experiments underlines the high DNA sequence and H3K9me3 specificity of BiAD sensor 4.

To detect locus-specific changes of H3K9me3 levels, we transfected the modules into the Suv39DKO iMEF cells, lacking both methyltransferase enzymes involved in setting pericentro-meric H3K9me3[52]. A strong 15-fold decrease in the number of

cells showing a specific BiFC signal was observed (Fig. 6c and Supplementary Fig. 24a). As a control, we used the 5mC-specific BiAD sensor 1, but did not observe differences between WT and Suv39DKO iMEFs, in line with an unaltered DNA methylation in the Suv39DKO iMEF cells (Supplementary Fig. 9a, b). Altogether, these results underscore the specificity of the novel BiAD sensor and indicate that this tool can be used to detect the reduction in H3K9me3 levels in a sequence-specific manner.

To detect an experimentally induced gain in H3K9me3 at these sites, we generated stable Suv39DKO iMEF cells, in which the expression of either WT or catalytically inactive (H324L) SUV39H1[53] is induced by addition of doxycycline (dox) (Supplementary Fig. 24b). Four days after induction, an increase in the global H3K9me3 levels was detected in total lysates obtained from WT but not H324L SUV39H1-expressing cells (Supplementary Fig. 24c). Remarkably, these changes in H3K9me3 levels could be traced in a locus-specific manner in live cells with the BiAD sensor 4, which revealed a tenfold increase in H3K9me3 signal 4 days after induction of the catalytically active SUV39H1 (Fig. 6b, c). In contrast, only a minor increase in BiFC signal was observed in the absence of dox (Fig. 6b) or after induction of the inactive SUV39H1 H324L variant (Fig. 6b, c). These data clearly demonstrate that the novel BiAD sensor can be used to specifically readout increasing H3K9me3 levels at major satellite repeats in live cells.

**Cell-cycle readout of 5mC levels with BiAD 1**. To follow changes in the levels of epigenetic marks in real time, we have generated NIH3T3 cells in which the modules of BiAD sensor 1 were genomically integrated and stably expressed (Supplementary Fig. 25a, b). In line with the heavy DNA methylation of pericentromeric heterochromatin, live-cell imaging of replicating cells revealed a strong BiFC signal on major satellites throughout the cell cycle (Fig. 7a). The detection of DNA methylation marks on mitotic centromeres agrees with the clustering of pericentromeric chromatin in mitosis and with published MBD localization data[12]. This result indicates that major satellite repeats are heavily methylated in all major stages of the cell cycle and that BiAD sensor 1 can be used to visualize the 5mC status of the target DNA sequences even when these are embedded in highly condensed mitotic chromatin structures. Moreover, the fact that the constant expression of the biosensor did not appear to perturb cell division documents the low cytotoxicity of this tool.

**Real-time readout of 5mC with BiAD 1 upon drug treatment**. To visualize drug-induced DNA methylation changes in real time, the cells that stably expresses the BiAD sensor 1 were imaged before 5-aza-dC treatment, 5 days after drug addition and 1 day after drug removal (Fig. 7b). Impressively, with this set-up we could observe a drastic drop in the percentage of cells displaying BiFC signals from circa 90% before 5-aza-dC treatment to around 6% on day 5 of the treatment (Fig. 7c). Imaging performed 24 h after drug removal revealed a rapid recovery of the BiAD fluorescence, with around 50% of the cells displaying BiFC signals. Altogether, these experiments highlight the applicability of stably expressing BiAD cell lines for the locus-specific tracking of DNA methylation changes in single cells.

## Discussion

The temporal order of epigenetic changes during the morphological and functional alterations of cells, as well as the dynamic connection between different epigenetic signals and changes in cellular physiology and morphology are key aspects that have remained mysterious so far. This is due to the lack of methods that enable a dynamic and locus-specific readout of epigenetic modifications inside the nucleus of living cells at the level of endogenous loci. In the present work, we addressed this urgent and unmet technological demand by developing several novel BiFC-based epigenetic biosensors for live-cell microscopy applications. With this toolbox, we were able for the first time to directly and specifically detect the status of 5mC and H3K9me3 signals at endogenous genomic sites. Furthermore, we could follow dynamic changes in these marks upon drug treatment, induction of epigenetic enzymes and during the cell cycle. As demonstrated, our technology facilitates the live-cell observation of epigenetic changes, thereby providing the possibility to directly correlate alterations in the epigenetic landscape with modifications in the cellular morphology and physiology.

In our work, several critical functional properties of BiAD sensors were studied. The fact that the BiFC signal is not dramatically altered as the cells progress throughout the cell cycle although their chromatin is massively reorganized from interphase to mitosis supports the notion that the BiFC signal arises from BiAD modules primarily binding next to each other at one genomic locus and not from association of the modules through 3D space. This is in agreement with results of chromosomal conformation studies documenting that genomic loci next to each other on the linear genome have a much higher probability to interact than more distant loci or loci located on different chromosomes[54]. In line with this, addition of trichostatin A

(TSA) to induce genome wide chromatin decondensation did not lead to significant changes in the intensity of the BiFC signals at the resolution of our measurements. Still, we cannot fully exclude that the BiFC signal partially arises from contacts formed in 3D through chromatin loops.

Our observation that BiAD sensor 1 remains associated with the mitotic apparatus as revealed by live-cell imaging with constitutively expressing cells, indicates a continuous binding of the sensor to chromatin. On the other hand, the finding that the cells are able to proceed through cell division suggests that the modules are able to transiently dissociate from chromatin during DNA replication.

To our knowledge, these novel sensors are the only tools available to date that enable visualization of epigenetic modifications with locus-specific resolution in the nucleus of living cells. Recently, another live-cell method was developed that uses the expression of a genomically inserted fluorescent reporter gene to measure the methylation state of the promoter adjacent DNA[55]. This method depends on the genetic modification of the target locus by nearby integration of a reporter gene and its dynamics is limited by the stability of the fluorophore. Moreover, it uses an indirect readout of the methylation state of the addressed locus by the monitoring the expression state of the inserted reporter gene. With the BiAD sensors developed in this study, in contrast, we achieved a direct, locus-specific readout of epigenetic marks at native endogenous genomic loci. Since the fluorescent signal is formed directly at the target locus, and it is not spread over the whole nucleus as in the approach of[55], the spatial information is preserved.

However, the binding of the BiAD modules might result in alterations of the local chromatin environment and this may in turn influence the BiFC signal. Furthermore, the perturbation of the "native" state of the system may be accentuated by the formation of the stable reconstituted fluorophore, which could act as a bridge between the binding sites of the anchor and detector modules and affect chromatin dynamics. Hence, future BiAD studies might be accompanied by analyses of DNA accessibility (like ATAC-seq) and chromatin structure (like as 3C or HiC-seq).

Although in this proof-of-concept study we focused on the readout of epigenetic marks at different types of repetitive DNA sequences, we demonstrate that by implementing CRISPR/dCas9 as an anchor module, the DNA-binding specificity of the sensors can be easily manipulated. In a recent study, Chen et al.[56] combined photoactivated localization microscopy with fluorescence complementation to detect protein complexes in live cells with nanometer resolution and single-molecule sensitivity. Using this microscopy technique should enable the design of tiled BiAD sensors for direct detection of epigenetic modifications at the level of single copy genes. This application would revolutionize our understanding of the dynamic properties of epigenetic signaling. Moreover, simultaneous readout of different epigenetic marks could be readily incorporated by making use of BiAD sensors with different colors or three-fragment fluorescence complementation systems[56]. By expanding the number of detector modules through incorporation of reading domains with specificities for other histone marks[57], the dynamics of bivalent chromatin domains could be assessed in live cells as well[58]. Moreover, since the BiAD approach is compatible with live-cell imaging, the relationship between locus-specific epigenetic modifications and cellular physiology can be directly addressed.

The ultimate aim of the BiAD approach developed in this work is to enable the real-time tracking of locus-specific epigenetic marks within the nucleus of single living cells, during cellular differentiation, pathogenesis or alterations in the cellular environment. For an improved real-time imaging of locus-specific epigenetic changes, the Venus fluorophore, for which the

reversibility of the complemented signal is still debated, could be replaced with IFP1.4[59–61], which was engineered for reversible fluorescence complementation[61]. Resorting to proteins capable of reversible fluorescence complementation could also help to reduce the effects that the BiAD sensors might have on the local chromatin environment at their target-binding sites and further improve the kinetic resolution of signal changes.

Based on the results obtained in our study, we envisage the generation of more model cell lines and of transgenic animals that stably express the BiAD sensors. This will allow the locus-specific detection of epigenetic changes during development, transgenerational epigenetics, cellular reprogramming, drug treatment, or onset of disease phenotypes. We anticipate that either in their current form or through combination with the recent developments in gene targeting and microscopy technologies, our tools will greatly contribute to a better understanding of how specific epigenetic signatures are set, erased and maintained at locus and cellular level, during normal cellular development and onset of disease.

## Methods

**Design of the BiAD plasmids.** Venus was used as reporter in the BiAD sensors due to its strong fluorescence intensity and its fast and efficient maturation properties[62, 63]. The 238 amino-acid protein was split at position 210 generating two non-fluorescent fragments (VenN and VenC). This selection was based on systematic experiments demonstrating a clear superiority of this split site over others in respect to its lack of nonspecific assembly, high specificity and signal intensity[24, 64].

The mVenus-C1 and mCerulean-C1 mammalian expression vectors were a gift from Prof. Steven Vogel[65] (Addgene plasmids no. 27794 and no. 27796). Cloning of the individual domains was performed using the Gibson assembly mix (New England Biolabs). All devices were N-terminally fused to a 3XFLAG tag for immunofluorescence and western blot detection. The domains were separated from the fluorophores through a flexible 14–18 aa linker. The synthetic anchor domains as well as the HP1 chromodomain were additionally tagged with the monopartite nuclear localization sequence (NLS) of the SV40 Large T-antigen for nuclear import. The methyl-binding domain of MBD1 contained an endogenous NLS. All fusion proteins were expressed under the control of a CMV promoter. The identity of all constructs was validated by sequencing. To generate the BiFC-based sensors, the Venus ORF was split at amino acid 210 and fused as described above to either the N- or C- terminus of the BiAD domains, through Gibson assembly. The sequences of all BiAD sensor plasmid are provided in the Supplementary Fig. 26.

**Cloning of the anchor domains.** The vector encoding for the GFP-fused ZF protein was provided by Dr Bert J. van der Zaal (Leiden University)[17]. The sequence encoding for the ZF was amplified by PCR and assembled in the BiAD module vectors as described above. The pTH-PanCen-mVenus was provided by Prof. Thoru Pederson (Addgene plasmid no. 49640)[18]. To avoid PCR artefacts due to the repetitive structure of the TALE gene, the template vector was digested with SbfI and XbaI (NEB) to release the gene of the fluorophore in the original vector. This was followed by Gibson assembly with PCR fragments encoding for mRuby2 (Dr Michael Davidson, Addgene plasmid no. 54768)[66], VenN or VenC. To increase the size of the linker that separates the TALE from the fluorophore from 7 to 18 amino acids, the first generation of BiAD modules was linearized with SbfI and assembled with an oligonucleotide cassette encoding for a GS-rich linker.

The dCas9-based anchor module was derived from the pHAGE-TO-dCas9-3XmCherry plasmid (provided by Dr Thoru Pederson, Addgene plasmid no. 64108)[19]. The vector was cut with BamHI and XbaI (NEB) to release the triple fluorophore fusion. The resulting fragment was then assembled with PCR inserts encoding for the VenN or VenC fragments, respectively. For targeting the alpha satellite repeats in the pericentromeric region of chromosome 9, the sgRNA against the 5′-TGGAATGGAATGGAATGGAA 3′ sequence was used as described by Ma et al.[19] (2015).

**Cloning of the detector domains.** To detect the 5mC mark, the methyl-binding domain of human MBD1 (accession Q9UIS9.2) was amplified out of HEK293 cDNA and cloned as described above. The construct borders were amino acid 1–113 as described[31]. The R44Q-binding pocket mutation, was introduced by site-directed mutagenesis PCR[67]. For the recognition of H3K9me3, the chromo-domain (amino acid 17–76 of mouse HP1β, accession NP_031648) was amplified from NIH3T3 cDNA and cloned into the BiAD sensor backbone as described above. The W42A-binding pocket mutation, was introduced by site-directed mutagenesis PCR[67].

**Cloning of epigenetic modification enzymes.** To rescue the methylation levels of the HCT116 DNMT1 hypomorphic cell line, a plasmid driving the expression of full length human DNMT1 (accession AAI26228.1) under CMV promoter was used. The catalytically inactive C1226A mutant version was employed as negative control[68]. Both plasmids were provided by Dr Pavel Bashtrykov (University of Stuttgart). The enzyme variants were fused to LSSmKate2 (Addgene plasmid no. 54795, provided by Dr Michael Davidson and Dr Vladislav Verkhusha) for microscopy-based detection. For cloning of the dox-inducible SUV39H1 expression plasmids, the CDS encoding for the full length mouse enzyme (accession NP_035644.1) was sub-cloned from a mammalian expression plasmid (provided by Dr Srikanth Kudithipudi, Stuttgart University) into the pSIN-TRE3G-PGK-Puro-IRES-rtTA3 vector[69]. Both the wild type and the catalytically inactive H324L variant, were cloned as mRuby2 fusions for microscopy-based detection.

**Cell lines.** HEK293 and NIH3T3 cells (American Type Culture Collection) were maintained in Dulbecco's modified Eagle's medium and high glucose (Sigma) supplemented with 10% heat-inactivated calf serum and 2 mM L-glutamine (Sigma). Wild-type and Suv39h1h2−/− iMEFs were a gift of Prof. Thomas Jenuwein (MPI Freiburg). p53−/− and p53−/−/Dnmt1−/− iMEFs were provided by Prof. Howard Cedar (Institute for Medical Research Israel-Canada). The cells were grown at 37 °C in Dulbecco's modified Eagle's medium and high glucose supplemented with 10% heat-inactivated calf serum, 1 × non-essential amino acids (Gibco), 1 × sodium pyruvate (Sigma), 0.1 mM β-mercaptoethanol (Gibco) and 2 mM L-glutamine. The wild-type and HCT116 DNMT1 hyphomorphic cells (kindly provided by Prof. Bert Vogelstein, HHMI, USA) and were cultivated in McCoy's 5 A medium (Gibco) supplemented with 10% heat-inactivated calf serum and 2 mM L-glutamine. All cells were grown at 37 °C in a saturated humidity atmosphere containing 5% CO₂.

**Inhibitor treatment.** To deplete HEK293 cells of DNA methylation, 5-aza-dC (Sigma-Aldrich, cat. no. A3656) treatment was performed over a period of 3 days at final drug concentration of 2 μM in the cell culture medium. The drug was dissolved in 50% acetic acid at 100 mM and replaced on a daily basis. As control, an equal volume of solvent, was added to the cell culture medium. To analyze the global efficiency of demethylation, total genomic DNA was isolated before, during and after the 5-aza-dC treatment using the DNeasy Blood and Tissue Kit (QIAGEN). Two-hundred nanogram of the resulting material was digested with the 5mC-inhibited enzyme HpaII (New England Biolabs). The DNA was resolved on a 0.8% agarose gel supplemented with GelRed (Genaxxon), and finally imaged with a Quantum imaging system (Vilber).

To determine the optimal TSA (Sigma-Aldrich, cat. no. T8852) concentration needed to increase the global histone acetylation levels with minimum cytotoxic effects, HEK293 cells were treated with 20, 80, and 330 nM TSA for 24 h. TSA was dissolved in dimethylsulphoxide (DMSO) at a concentration of 5 mM. Afterwards, cells were harvested and lysed for 30 min on ice in lysis buffer (20 mM HEPES pH 7, 500 mM NaCl, 0.5% NP-40, 2.5 mM MgCl₂, and 0.2 mM phenylmethylsulphonyl fluoride) followed by sonication with EpiShear (Active Motif) for 45 s (15 s ON, 30 s OFF cycles, 40% power, 1/8″ microtip) to release nuclear proteins. The lysate was next centrifuged at 15,000xg, 4 °C, 15 min and the resulting supernatant, containing the nuclear fraction, was analyzed by western blotting. To assess the global histone acetylation levels, an anti-H4panAc antibody was used (Active Motif, cat. no.530804, lot no. 530804). This was followed by incubation of the nitrocellulose membrane with an anti-rabbit horseradish peroxidase (HRP)-coupled secondary antibody (GE Healthcare Life Sciences, cat no. NA934V, lot no. 9670531). The chemiluminescence signal was detected on a FUSION Solo (Peqlab) system. For microscopy, 330 nM TSA were added 24 h before the slides were fixed. Equal volumes of the DMSO carrier were added to the control cells.

**Amplicon-targeted bisulfite sequencing.** To determine the methylation levels of the loci adjacent to th1e TALE-binding sites (TBSs) in the HCT116 DNMT1 hypomorphic cells, the hg38 reference genome was mined for TBSs. 720-bp long sequences centered around the 20-bp long TALE-binding motif were extracted from all TBSs and ranked based on their CpG content. With this approach, the 57999639: 58000359 locus on chromosome 7 was among the sequences with the highest number of CpG sites (12 within the PCR amplicon) and it was selected for DNA methylation analysis (Supplementary Fig. 15a).

For Illumina library preparation, genomic DNA was extracted from HEK293, HCT116, and HCT116 DNMT1 hyphomorphic cells using the QIAamp DNA Mini Kit (Qiagen), and bisulfite converted using EZ DNA Methylation-Lightning Kit (Zymo Research) following the manufacturer's instructions. The bisulfite treated DNA was then used for PCR amplification using the amplicon and primers with cell-type specific barcodes (listed in Supplementary Table 8) and HotStarTaq DNA Polymerase (Qiagen). PCR products were resolved on an 8% acrylamide gel, followed by extraction and clean-up using the NucleoSpin Gel and PCR Clean-up (Macherey-Nagel). The products were mixed at an equimolar ratio and sent for paired-end sequencing on Illumina HiSeq2000 to Novogene Bioinformatics Technology Co., Ltd., Beijing, China (www.novogene.cn).

The high-throughput sequencing results were demultiplexed and analyzed using the CLC Genomic Workbench 10.0.1 (CLC Bios, MA) following the manufacturer's standard data import protocol and the bisulfite sequencing plugin. The reads were mapped to the Chr7 amplicon reference sequence with the length and similarity fraction parameters set to 0.85. Over 99% of the reads could be successfully mapped to the reference sequence using this approach. The methylation levels normalized to the total read number where then extracted for each CpG site of the amplicon, based on at least 1600 reads in each cell line. Further visualization and analysis was performed in MS Excel.

**Western blot of BiAD module expression.** To detect the expression of the BiAD modules, HEK293 cells were transiently transfected with Fugene HD (Promega) according to the manufacturer's recommendations. Twenty-four hours after transfection, the cells were harvested and lysed as described above. To assess the impact of the R44Q mutation on the stability of the MBD domain, the mCerulean plasmid was co-transfected together with the detector module. This was used to compare the expression levels of the two MBD variants, independent of variabilities in transfection efficiency. For evaluating the expression levels of the BiAD modules, the nuclear lysate was separated on a 15% sodium dodecyl sulfate polyacrylamide gel electrophoresis (SDS-PAGE) followed by transferring onto a nitrocellulose membrane. For detection, the monoclonal anti-FLAG M2 (Sigma-Aldrich, cat. no. F1804, lot no. SLBN5629V, 1:1000 dilution) antibody was used. This was followed by incubation with a HRP-coupled mouse secondary antibody (GE Healthcare Life Sciences, cat. no. NA931V, lot no. 9653127, 1:5000 dilution). For mCerulean detection, the rabbit Living Colors full-length GFP (Clontech, cat. no. 632592, lot no. 1404005, 1:1000 dilution) antibody was used. Uncropped blots are provided in Supplementary Fig. 27.

**Immunofluorescence.** To determine the cellular localization of the BiAD modules in HEK293 and NIH3T3 cells, cells were seeded on microscopy slides and transfected with the corresponding plasmids, using Fugene HD. 24–48 h after transfection, the slides were washed with MgCl$_2$ and CaCl$_2$ containing PBS (Sigma-Aldrich), followed by crosslinking with 4% formaldehyde solution (Sigma-Aldrich) for 10 min at room temperature. This was followed by permeabilization with 0.5% Triton X-100 (Sigma-Aldrich) for 10 min at 4 °C and blocking with 1% BSA solution pH 7.5 for 1 h at room temperature (Sigma-Aldrich, lot no. SLBN5629V). The mouse anti-FLAG M2 and rabbit anti-H3K9me3 (Active Motif, no. 39161 lot no. 13509002, 1:500 dilution) primary antibodies were used. For microscopy-based detection, the corresponding Alexa Fluor 594-conjugated anti-mouse or anti-rabbit (Invitrogen, cat. no. A-11062 and A-11037; lot no. 49401 and 56948 A, respectively, 1:1000 dilution) secondary antibodies were used. For nuclear staining, a 5 min incubation step with 1 μg mL$^{-1}$ DAPI (Thermo Fisher Scientific) was included before mounting.

**BiFC assay.** For the BiFC assays, cells were seeded on high precision no.1.5 (tol. ± 5 μm) glass slides (Carl Zeiss) and transfected with the BiAD modules as indicated in Supplementary Tables 1–4. Depending on the cell line and the sensor to be transfected, following transfection reagents were used: Fugene HD (Promega), GenaxxonFect (Genaxxon), and Lipofectamine 3000 (Thermo Fisher Scientific) following the recommendations of the supplier. NLS-mRuby2-C1 was routinely used as a transfection control due to the absence of crosstalk in the BiFC channel. In experiments where red fluorophore-tagged epigenetic enzymes were co-expressed with the sensors, mCerulean-C1 was used as a transfection marker. To keep the ratio between the total DNA and transfection reagent constant, corresponding amounts of pcDNA3.1 (Invitrogen) were used as carrier. For validating the targeting specificity of the sensors, co-transfections with the full fluorophore-tagged modules was performed. These plasmids were transfected at half of the amount as to what was used to split fluorophore-fusions, to compensate for the difference in fluorescence intensity. For studying the dynamics of epigenetic changes, the cell treatment was performed 24–48 h before transfection with the BiAD modules. For the 5-aza-dC methylation kinetics, the BiAD sensor was transfected 48 h before every imaging time-point. The time scale annotated in Fig. 3 represents the number of days since the start of the 5-aza-dC treatment and since recovery time, respectively. For imaging, the slides were fixed for 10 min at room temperature with 4% formaldehyde and finally mounted in ProLong Diamond antifade (Invitrogen).

The slides were imaged on an LSM 710 Zeiss confocal microscope equipped with a Plan-Apochromat 63 × /1.40 Oil DIC M27 objective. The laser excitation wavelengths as well as emission collection windows are indicated in Supplementary Table 5. For enhanced sensitivity, the BiFC channel signal was routinely directed to a QUASAR 34-channel photomultiplier unit (Carl Zeiss). Image analysis was performed in ImageJ 1.51a. To account for the reduced brightness of the BiAD sensor 4, the intensity of the 514 nm laser line was increased to 5%.

For analysis, the BiFC reconstitution pattern was broadly categorized by visual inspection into three classes: strong spotty, weak spotty/blurry, and no signal. To reliably identify transfected cells without detectable BiFC signal, NLS-mRuby2 was routinely included as a transfection control. To account for the imbalances in plasmid expression between the BiAD modules and the transfection control, the cells were first visualized in the red channel. Only the cells that showed a detectable red fluorescence signal were used for BiFC signal analysis. For the BiAD sensor 4,

dCas9-3xmCherry was used as an internal transfection and DNA sequence specificity control. For the quantification of the BiFC signal observed in the cell line with stable integration of the BiAD 1 sensor modules, the cells were split into two categories only: BiFC positive and BiFC negative. This was based on the lack of the BiFC signal in the cells expressing the MBD R44Q variant, indicating that the BiFC fluorescence detected for the MBD WT cells is specific and not due to overexpression conditions, as it might be the case in transient transfections. In experiments comparing fluorescence intensities between different conditions, transfections of the constructs to be compared as well as their imaging were performed in parallel using the same settings. All experiments were performed in duplicates and 15–60 cells were counted per biological replicate or, where applicable, per time point. The sample size was selected to ensure a proper representation of the three BiFC patterns, namely strong spotty, weak spotty/blurry, and no signal, within each transfection set-up. The error bars represent standard error of the mean. Further details are provided in Supplementary Table 6. p-values were determined for the "strong spotty" category by performing 1-sided paired t-test analysis in MS Excel (see Supplementary Table 7).

**Implementation of the Tet-SUV39H1-mRuby2 dox-inducible iMEFs.** To generate a suitable cellular system for assessing the dynamic range of the BiAD sensor 4, doxycycline-inducible cells lines were created by retroviral infection of the parental Suv39DKO iMEFs, with viruses delivering transgenes encoding for either the SUV39H1 or the catalytically inactive H342L mutant both fused to mRuby2. Retroviral packaging was performed using Platinium-E cells (Cell Biolabs) according to established protocols[69]. In brief, for each calcium phosphate transfection, 10–20 μg plasmid DNA and 5 μg helper plasmid (pCMV-Gag-Pol, Cell Biolabs) were used. Transduced Suv39DKO iMEFs were selected 48 h after infection using 1 μg mL$^{-1}$ puromycin. Two weeks after selection, protein expression was induced with 1 μg mL$^{-1}$ doxycycline, final concentration in media. The successful expression of SUV39H1-mRuby2 was confirmed by fluorescence microscopy. Both protein variants displayed comparable expression levels, as judged by their fluorescence intensity. To validate the activity of the constructs, samples cells were collected before, during and after dox induction, at the indicated time points. Equal volumes of cell lysates prepared as described above, were loaded on a 18% SDS-PAGE and subjected to western blot using anti-H3K9me3 and anti-β actin (Abcam, cat. no. ab8227, lot no. GR576921, 1:2000 dilution) antibodies. To test the applicability of the BiAD sensor 4, the expression of the SUV39H1 variants was induced with 1 μg mL$^{-1}$ doxycycline, 48 h before transfecting the cells with the biosensor modules. Cells were cultured in the presence of dox for another 48 h prior fixation for microscopy imaging.

**Generation of cell lines with stable expression of BiAD 1.** To generate cell lines with stable expression of the BiAD 1 modules, the MBD-VenC and ZF-VenN modules were sub-cloned by Gibson assembly into a modified version of the pMSCV-LTR-miRE-PGK-Puro-IRES-GFP vector giving rise to the pMSCV-LTR-MBD_VenC-IRES-ZF_VenN-PGK-Puro construct. Two vectors were created with this approach, whereby one contained the WT MBD domain, while in the second this was replaced with the MBD R44Q variant. Following sequencing, these vectors were used for virus production and infection of NIH3T3 cells, following the procedure described above. Transduced cells were selected 48 h after infection using 1 μg mL$^{-1}$ puromycin. The successful expression of the MBD–VenC and ZF-VenN modules was assessed by immunofluorescence and western blotting with the anti-FLAG M2 antibody (Sigma-Aldrich, cat. no. F1804, lot no. SLBN5629V), following the procedures described above. β-actin detection (Abcam, cat. no. ab8227, lot no. GR576921) was used to account for differences in the amounts of loaded lysates.

**Live-cell imaging.** For live-cell imaging, cells were seeded on 35-mm Fluorodish cell culture dishes (World Precision Instruments). Growth media without phenol red was used and imaging was performed on an LSM 710 Zeiss confocal microscope equipped with a Plan-Apochromat 63 × /1.40 Oil DIC M27 objective and an XL-LSM 710 S1 incubation chamber for temperature and CO$_2$ control. In transient transfection experiments with the sensors, transfection conditions, image acquisition, and display settings were identical to what was used for formaldehyde-fixed cells to allow the direct comparison of the observed BiFC signals. To minimize laser-induced phototoxicity and accommodate for differences in BiFC signal intensities between cells with transient and stable expression of the BiAD modules, the thickness of the imaged slice was enlarged from 0.8 to 1.6 μm for the cell-cycle imaging of the BiFC signal. Time-lapse imaging was performed for 18 h, with images being taken every 25 min. A representative subset of time-points was selected for display in Fig. 7a. Image acquisition and display settings were identical within the panels of the figure.

For visualization of drug-induced changes of DNA methylation, NIH3T3 cells with stable expression of the BiAD modules were subjected to 5-aza-dC (Sigma-Aldrich, cat. no. A3656) treatment. This was performed over a period of 5 days at a final drug concentration of 0.2 μM in the cell culture medium. Images were taken before, at the end of the treatment and 1 day after 5-aza-dC removal. Image acquisition and display settings were identical between the time-points. To facilitate data visualization, fluorescence intensities are presented as a look-up table from dark purple to bright yellow (fire LUT) generated with ImageJ 1.51a.

**Data availability**. Bisulfite sequencing data has been deposited at Sequence Read Archive (SRA) under accession code SRP111502. All key data supporting the findings of this study are available within the paper and its supplementary information files. Additional primary data are available from the corresponding author upon reasonable request.

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

## Acknowledgements

C.L. has been supported by a PhD fellowship provided by the Carl Zeiss foundation. We are very grateful to the Central Facility for Advanced Microscopy of the Stuttgart Research Center Systems Biology at the University of Stuttgart for providing access to the laser scanning microscope. We are thankful to Dr Stephan Eisler, Max Emperle, Dr Pavel Bashtrykov, Dr Renata Jurkowska and Dr Srikanth Kudithipudi for technical support and discussions.

## Author contributions

C.L. and A.J. designed the research. C.L. and S.P. performed the experimental work. P.R. provided advice and materials for the generation of the dox-inducible cell lines. J.B. performed the bioinformatic search for TALE binding sites. C.L. and A.J. wrote the manuscript draft. All authors were involved in data analysis and finalizing of the manuscript.

## Additional information

**Competing interests:** The authors declare no competing financial interests.

