## [Peer Review file · Nature Communications]

Reviewers' comments:

Reviewer #1 (Remarks to the Author):

The authors describe the creation of biosensors for epigenetic marks using fusion of non-fluorescent complementary fragments of a fluorescent protein to a DNA-binding protein and an epigenetic mark "reader" domain, respectively. This is an important advance for monitoring chromatin state in live cells. They apply this new live-cell fluorescence technology to detecting H3K9 methylation with the chromodomain of HP1-beta and DNA methylation in mouse and human cell lines with the 5mC binding domain of MBD1. The authors show that BiAD sensor detection depends on sequence specificity and reader domain activity using mutant controls with the overall appearance of the fluorescence pattern as the readout. They test one sensor for methylation a specific locus on Chromosome 9 and detect the expected number of localized spots. They check the performance of the BiAD sensors in a time course using inhibition of DNA methylation and recovery. The sensor's ability to detect dynamic changes in methylation is very important for its usefulness to the field, and this experiment enhances the impact of the paper. Overall, this paper presents an important new tool for monitoring chromatin state in live cells and use of reader domain mutants as controls for specificity is good. However, the paper suffers from a lack of quantitative detail in evaluating the fluorescence images. Few cells (only 20-30) are quantified per replicate and per condition, and the quantitation is done manually. Quantitative changes in chromatin structure or subtle artifacts may not be detected. Furthermore, there appears there may be a lack of concordance between the transfection control image and the BiAD sensor image, which makes it difficult to interpret. Because fluorescence levels are so critical to the interpretation of this paper, the authors should be more clear in describing imaging conditions (which panels were matched, vs. different exposures) and contrast manipulation for each figure. If feasible, quantifying larger numbers of cells and using an automated image analysis software to classify and quantify the fluorescence pattern of the BiAD sensor is recommended to make the data more convincing. Furthermore, targeting is only evaluated using microscopy, without the use of orthogonal methods like ChIP-qPCR -- whenever a molecular change is expected to be seen, it should be validated with an orthogonal methods to demonstrate that it has actually occurred. A more explicit and detailed discussion of potential changes in 3-D chromatin contacts due to binding of the split FPs, with appropriate controls such as 3C experiments, would also be welcome.

Comments and questions about the manuscript

Fig. 1 b,d: There is a dramatic difference in the nuclear/cytoplasmic distribution of transfection control (mRuby?) between panel b and panel d. Is this representative of normal diversity in distribution of the transfection control or has the contrast been changed? This should be addressed because the representative image is so different. Caption should indicate whether the contrast/exposure conditions are the same between control and experimental panels.

Fig. 1 e: Is there only one replicate in the Dnmt1 -/- cells? There is no error bar shown. Also, 20-30 cells per replicate is low, given the fact that there are 3 categories. For example, in the iMEF Dnmt1 -/- sample, if there is just one replicate, then 10% means that bar only represents 2-3 cells.

Do the authors show co-localization on the same stretch of DNA or merely in the same general 3D region where the two targets for detection (particular sequence and histone modification or DNA methylation) could diffuse into each other transiently? Figure S10 is helpful, showing that BiAD detection of methylation does not change in the Suv39DKO cells, but not fully convincing. The binding between the two halves of the fluorescent reporter could stabilize a transient interaction, potentially creating a broad area of false negative detection, or at least detection of weak 3D co-localization rather than strong 1D co-occurrence at the same locus on the chromosome. It is also not proven that the sensor is associating on DNA rather than in a general nuclear region surrounding the target loci due to increased local concentration through weak interactions. Although the latter is still useful for detecting weak interactions, ChIP-qPCR at several loci if

interest could be used as an orthogonal method to show convincingly that the two parts of the BiAD sensor are associating with the DNA at a level that can be detected by crosslinking. In any case, whether the sensor binds tightly to DNA or only weakly probes it should be discussed. In general, orthogonal "true positive" data (orthogonal genomic methods) need to be presented to ensure these experiments are working the way they are expected to work.

The evaluation of BiAD Sensor 2 could be more convincing if a pericentromere or centromere-associated protein was imaged by immunofluorescence at the same time as detecting the spotty BiFC signal – do they actually colocalize?

Interactions between the split FPs could alter the association of genomic loci in living cells. The authors should discuss this potential source of artefacts in more detail. One way to discuss this would be using comparisons of BiFC signal and colocalization microscopy as shown in Fig. S3. Artefactual chromatin associations caused by the split FP would be subtle, so a quantitative analysis of the spot sizes and signal distribution for colocalization microscopy and BiFC could be helpful to evaluate whether this is a problem.

It should be checked if the split FPs artefactually bridge chromosomal loci together, changing chromatin conformation in the cell. For single locus targeted sensors (such as the one on chromosome 9), 3C could be used to validate that 3-D chromosomal contacts don't appreciably change in the presence of the sensor.

How much time passed between transfection and imaging? There is only information on this for the 5-aza-dC experiment and the iMEF experiment in the methods (p. 19). For those, it looks like the time between transfection and imaging was 48 hours, which is longer than recommended (Kerppola, Nature Protocols, 2006). Why was such a long time point used, and could it lead to nonspecific signal?

Figure S4a, top two panels: The strong BiFC signal does not seem to colocalize with cells that have strong transfection control signal, so the figure panel does not support what is asserted in the caption. Many nuclei have BiFC signal where there is no visible signal in the transfection control channel. Can the authors elaborate on this, and maybe use thin white outlines to mark the cells that have strong transfection to clarify? Quantifying the total transfection control signal and total BiFC signal in each cell for a larger number of cells would also make this more interpretable.

Figure 3c: It is particularly important here to describe whether the contrast is the same in all panels.

Figure S9c: Two cells per condition are not sufficient to evaluate if BiFC signal changed during treatment with TSA. Please include more images or a quantification as in Figure 3.

Minor issues

p. 3, 2nd paragraph: "has not be achieved" should read "has not been achieved"

p. 3, 2nd paragraph: Non-affinity-based methods like bisulfite sequencing and methylation-sensitive PCR are also used to detect 5mC

p. 3, last paragraph: several awkward phrasing and a typo "mintbodies"

p. 4, 2nd paragraph: What motivated the choice of the HP1-beta chromodomain?

p. 4, 3rd paragraph: acronym BiFC is not defined.

Fig. 1 caption, b,d: Fluorescence channel shown in red is not explained.

p. 6, 3rd paragraph: Typo in "to proof its 5mC specificity"

p. 9, 2nd paragraph "interested to explore" awkward phrasing

p. 20: formatting error in reference 1: "Nature reviews. Genetics". Same formatting issue in references 4 and 9.

p. 23, ref. 44: The authors should cite the original work where the 7 nm distance is calculated or estimated, not a paper that cites other work for this number. From a brief look at the paper cited

and the citations within that paper, it is not clear how the 7 nm distance is calculated.

Overall, standard error on the mean of 2 measurements should not be reported, just plot the results of the two measurements.

Reviewer #2 (Remarks to the Author):

Modular fluorescence complementation sensors for live cell detection of epigenetic signals at endogenous genomic sites

Lungu et al

This manuscript presents an interesting approach to visualize the epigenetic status of specific DNA loci. This method relies on fusing a split fluorescent protein into two sensors, one detecting the DNA or Histone modification and the other detecting a specific DNA sequence. Upon binding of the sensors in close proximity the full fluorescent protein is reconstituted and signal can be detected. The authors present two "epigenetic" sensors: one detecting DNA methylation and one detecting the H3K9me3 modification. We recognize the innovative component of this system and its potential for live imaging and tracking of epigenetic modification. In addition, we appreciate the fact that this method is able to detect changes in the enrichment of these epigenetic modifications. However, as presented the method still falls short of its main goals.

The advantage of this system compared to currently existing methods (such as PLA) is the ability to track changes that are occurring in living cells. However, the authors do not show any videos of their experiments. How long is it possible to track these cells for? Can the cells be followed through cell division? Is the signal lost during mitosis? The authors should show videos and representative images of cells over one to two days. This should be done for both DNA methylation and H3K9me3.

As the authors point out in their discussion this work is still limited to visualization of large repetitive regions. As such, almost all figures shown here could have been done with the sensors that detect the epigenetic modifications. The only two figures that have some specificity at the level of DNA loci are 2D and 3F where the authors use dCAS9 to label pericentromeric repeats on chromosome 19. The authors should show that their system is able to detect a single locus. This could be done by targeting single loci that contain several repeats that could be bound by a single gRNA (like the MUC or IGH genes) but also by targeting several sgRNAs to one location. Unless the method really is able to allow visualization of specific single loci (instead of repeats) we do not believe it will be useful for the community in a way that grants publication in Nature Communications.

Could the authors demonstrate that the system works with active histone marks like H3K27ac or H3K36me3 – use of this system for tracking an actively transcribing gene is important.

Finally, the intensity of the signals shown in figure 4a for the H3K9me3 modification is a bit worrisome. The signal to noise ratio seems very low and cells seem to have been exposed for quite some time. If that is the case does it become toxic?

Other points:

All figures where DAPI is used to validate a finding should be shown with the merged channels to allow proper evaluation. In general all figures should have a merged version.

All the figure callouts are in totally random order – this is extremely annoying. Figures should be reorganized to follow the text.

Reply to the reviewers' comments

General comments

We like to thank the reviewers for working with our manuscript and their insightful and constructive comments which helped us to further improve the content and its presentation. In addition to changes in the structure of the manuscript and clarifications in the line of argumentation, we have performed new series of experiments that convincingly show that the BiAD sensors can be used to specifically visualize changes of target epigenetic modifications in living cells, with locus specific resolution. These include the following:

1. In response to reviewer #1 who was criticizing the quantitative assessment of the performance of our BiFC sensors, we have now included one extra biological replicate for experiments shown in the following panels: new figures 5f, Supplementary Figs. 9b and 17d. In addition, we have calculated p-values for the 'strong spotty' category of each figure (Supplementary Table 7), which were very favorable.
2. To complement point 1) we have also established and validated a pocket mutant variant of the HP1CD, which is no longer able to recognize H3K9me3 modifications. Implementing this as negative control in BiAD sensor 4 resulted in a dramatic reduction in the number of cells showing BiFC signals, highlighting the specificity of our sensor (new Supplementary Fig. 23).
3. In response to reviewer #1 who was asking for orthogonal methods that validate the results obtained with the BiAD sensors, we have performed amplicon-targeted bisulfite sequencing to orthogonally validate that the TALE binding sites have reduced methylation in the DNMT1 hypomorphic HCT116 cell line (new Supplementary Fig. 15). In line with the changes in the BiFC signals, we observed a 40% drop in DNA methylation levels in these cell lines. In addition, we like to mention that the anchor domains were taken from published data where they already have been validated by several methods. This is now more explicitly stated in the revised manuscript.
4. In response to reviewer #1 who raised the possibility that the BiFC signal might result from the association of the BiAD modules in 3D and not 1D on chromatin, we have performed a quantitative assessment of the BiFC signal in TSA-treated cells. This drug was shown to induce local chromatin decondensation and is expected to interfere with the BiFC signal if this arises through 3D contacts. No significant changes between mock and TSA-treated cells were observed, supporting the 1D association of the modules (new Supplementary Fig. 17d).
5. In response to reviewer #2 who was criticizing that in our study we do not track the BiFC signals in living cells, we have now included representative images recorded in living cells transfected with BiAD sensor 1 (new Supplementary Fig. 3). The high mark and sequence specificity of the sensor was preserved under these imaging conditions.
6. In response to reviewer #2 who was asking for long-time imaging of the cells expressing the BiAD sensors, we have generated and validated cell lines with genomic integration and stable expression of the BiAD sensor 1 (new Supplementary Fig. 25). With these, we could follow in real time, the changes in the BiFC signal during the cell cycle, as well as during 5-aza-dC treatment (new figure 7). Both experiments were successful and clearly document the dynamic imaging of epigenetic marks with locus resolution in live cells by BiAD as requested by the reviewer.

Please find below our detailed reply (printed in blue) to the reviewers' comments.

Reply to Reviewer 1

“The authors describe the creation of biosensors for epigenetic marks using fusion of non-fluorescent complementary fragments of a fluorescent protein to a DNA-binding protein and an epigenetic mark “reader” domain, respectively. This is an important advance for monitoring chromatin state in live cells. They apply this new live-cell fluorescence technology to detecting H3K9 methylation with the chromodomain of HP1-beta and DNA methylation in mouse and human cell lines with the 5mC binding domain of MBD1. The authors show that BiAD sensor detection depends on sequence specificity and reader domain activity using mutant controls with the overall appearance of the fluorescence pattern as the readout. They test one sensor for methylation a specific locus on Chromosome 9 and detect the expected number of localized spots. They check the performance of the BiAD sensors in a time course using inhibition of DNA methylation and recovery. The sensor’s ability to detect dynamic changes in methylation is very important for its usefulness to the field, and this experiment enhances the impact of the paper. Overall, this paper presents an important new tool for monitoring chromatin state in live cells and use of reader domain mutants as controls for specificity is good.”

Reply: Thank you very much for these positive words.

“However, the paper suffers from a lack of quantitative detail in evaluating the fluorescence images. Few cells (only 20-30) are quantified per replicate and per condition, and the quantitation is done manually. Quantitative changes in chromatin structure or subtle artifacts may not be detected.”

Reply: The focus of this work was to demonstrate the application of the BiAD sensors, which was successful in every single case. Since the differences between the compared conditions were very clear in all case, we think that 15-60 cells per biological replicate and per condition were sufficient. To further support our conclusions, we have calculated p-values for each figure, which are very favorable indeed (Supplementary Table 7).

“Furthermore, there appears there may be a lack of concordance between the transfection control image and the BiAD sensor image, which makes it difficult to interpret. Because fluorescence levels are so critical to the interpretation of this paper, the authors should be more clear in describing imaging conditions (which panels were matched, vs. different exposures) and contrast manipulation for each figure. “

Reply: We agree that for the proper interpretation of the BiAD signals, fluorescence levels are key data. We have now included a detailed description of the imaging conditions in the methods section and legends of the figures. Technically, we believe that there are three aspects that have to be addressed here.

First, we would like to draw the attention of the reviewer to the fact that the sensors and the accompanying transfection control plasmids used in transient transfection experiments were expressed from independent plasmids. This can give rise to variability in expression between the different fusion proteins, an issue already known in the field. (Grefen & Blatt, *BioTechniques*, Vol. 53, No. 5, November 2012, pp. 311–314). Since we were aware of this limitation, we purposefully did not attempt to correlate the intensity of the transfection control with the intensity of the BiFC signals, as this may lead to inconclusive results. Instead, the transfection control was interpreted as a yes/no signal. Practically, this means that only when

a cell showed a detectable Ruby signal, its associated BiFC signal was taken into consideration. This approach has now been clarified in the methods section.

A second limitation of transient transfections with separate plasmids comes from the high heterogeneity in fluorescence signal distribution. We have selected the imaging conditions such that fluorescent signals are not oversaturated. However, with this approach, weak fluorescent cells might appear as non-fluorescent. Correlated with the asymmetric distribution of plasmids in transient transfection experiments we agree that this may occasionally give rise to cells with no visible transfection control expression, but clear BiFC signal. To clarify this point we have included a revised version of Supplementary Fig. 4, where a zoom-in part of the main figure is displayed with enhanced brightness, to show that cells displaying BiFC signal do show mRuby2 fluorescence, albeit weak.

Finally, regarding data acquisition, all experiments that were to be compared directly were performed in parallel to account for potential handling issues. Data acquisition was performed in the same imaging session to average out potential artifacts coming from the laser lifetime and fluctuations in the laser lines. Contrast enhancements were performed with the same settings for all experimental sets that were to be compared directly and such that all cellular features were maintained during the process. This is now explicitly described in the methods section and in the figure legends.

“If feasible, quantifying larger numbers of cells and using an automated image analysis software to classify and quantify the fluorescence pattern of the BiAD sensor is recommended to make the data more convincing.”

Reply: We agree that this would be a very useful further development of the system in future biological applications. However, as detailed above, the unambiguousness of the results, does not make application of these additional tools or increase of the statistical basis necessary. As described above, we have now calculated p values for all the experiments where different setups are compared. In addition, we have included one extra biological replicate for experiments shown in the following panels: Figure 5f and Supplementary Figs. 9b and 17d.

“Furthermore, targeting is only evaluated using microscopy, without the use of orthogonal methods like ChIP-qPCR -- whenever a molecular change is expected to be seen, it should be validated with an orthogonal methods to demonstrate that it has actually occurred. A more explicit and detailed discussion of potential changes in 3-D chromatin contacts due to binding of the split FPs, with appropriate controls such as 3C experiments, would also be welcome.”

Reply: The binding of all DNA binding devices have been extensively validated in the original papers where they were introduced. This now has been mentioned more explicitly in the revised manuscript. While we initially considered ChIP-qPCR to validate that the signal reconstitution takes place at the targeted sequences, a more careful consideration lead us to conclude that in a BiFC approach this is technically not possible as one would need to ChIP the fully reconstituted fluorophore very specifically for meaningful data interpretation. We are not aware of any antibody that selectively recognizes the reconstituted Venus fluorophore. By using currently available antibodies one would ChIP the not reconstituted large module of the BiAD sensor as well, which would not allow to conclude if both modules are simultaneously present on the same site. Alternatively, one could tag the anchor and detector modules with different tags and perform re-ChIP. However, this will just show that the anchor and detector modules target the same genomic locus (concluded from microscopy studies as well), but not if that their co-binding leads to productive BiFC signals. Based on this reasoning we have

resorted mainly to microscopy techniques to validate our sensors (e.g. co-transfection with MBD Cerulean or dCas9-3*mCherry).

At the suggestion of the reviewer we have however performed amplicon-targeted bisulfite sequencing to orthogonally validate that the TALE binding sites have a reduced methylation in the hypomorphic DNMT1 HCT116 cell line (new Supplementary Fig.15). Like for the signal produced by the BiAD sensor 2, we observe also in this case a reduction in the levels of DNA methylation. In addition, we have also generated a mutant in the binding pocket of HP1 (HP1CD W42A). As documented in the new Supplementary Fig.23, the implementation of this detector module variant resulted in a strong drop in BiFC signals, further highlighting that BiAD sensor 4 requires an intact HP1CD for signal reconstitution to occur.

The proposal to discuss potential 3D change in chromatin after expression of the sensors is excellent and the ideas to use 3C as control is very good as well. Both points have been added to the discussion section.

“Comments and questions about the manuscript

Fig. 1 b,d: There is a dramatic difference in the nuclear/cytoplasmic distribution of transfection control (mRuby?) between panel b and panel d. Is this representative of normal diversity in distribution of the transfection control or has the contrast been changed? This should be addressed because the representative image is so different. Caption should indicate whether the contrast/exposure conditions are the same between control and experimental panels.”

Reply: We would like to thank the reviewer for picking up this point of potential confusion. mRuby-NLS and not mRuby was used as a transfection control and as a marker for the nucleus. This has now been corrected in the methods and we like to apologize for this mistake in the original manuscript. For the negative control panels, we have purposefully selected cells with a higher mRuby2-NLS signal to show that, keeping in mind the caveats described above, even under high expression of the modules, there is no detectable BiFC signal. Since the size of mRuby-NLS is smaller than that of the nuclear pore, the protein can diffuse passively in the cytoplasm despite the presence of the NLS. This is particularly pronounced in stronger expressing cells. Since for the wild-type setup, we selected a weaker transfected example cell, here the fluorophore still maintains its nuclear distribution. To avoid confusions we have exchanged this panel such that the transfection control has a comparable localization between the different experimental setups. As described above, acquisition and contrast conditions are same between the control and experimental panels. This is now explicitly stated in the figure legends.

“Fig. 1 e: Is there only one replicate in the Dnmt1 -/- cells? There is no error bar shown. Also, 20-30 cells per replicate is low, given the fact that there are 3 categories. For example, in the iMEF Dnmt1 -/- sample, if there is just one replicate, then 10% means that bar only represents 2-3 cells.”

Reply: We would like to thank the reviewer for picking up the fact that the error bar in this category was missing. This was inadvertently omitted and has now been corrected. Please note as mentioned above, that the experiments were performed in minimum duplicates and that results of 30-60 cells are combined in each panel.

“Do the authors show co-localization on the same stretch of DNA or merely in the same general 3D region where the two targets for detection (particular sequence and histone modification or DNA methylation) could diffuse into each other transiently? Figure S10 is

helpful, showing that BiAD detection of methylation does not change in the Suv39DKO cells, but not fully convincing. The binding between the two halves of the fluorescent reporter could stabilize a transient interaction, potentially creating a broad area of false negative detection, or at least detection of weak 3D co-localization rather than strong 1D co-occurrence at the same locus on the chromosome.”

Reply: The efficiency of fluorophore reconstitution is directly related to the overlap of the density distributions of both parts. We agree with the reviewer that there is a hypothetical possibility of detecting 3D instead of 1D contacts. However, as clearly demonstrated in chromatin conformation experiments, strong local contacts (in cis) are highly preferred over 3D contacts (in trans) due to the dynamic nature of the chromatin fiber.

Furthermore, apart from the experiment mentioned by the reviewer, there are two other experimental approaches included in our work where we address this point and the results of which favor the 1D vs the 3D association. First, TSA treatment was performed since this was reported by Ricci et al, Cell, 2015 to lead to a local opening of the chromatin fiber. We have included the quantification of the BiFC signal from cells expressing BiAD sensor 2 and that have been either mock or TSA treated (new Supplementary Fig. 17) and observed no significant changes in the distribution of BiFC patterns between the two conditions ($p > 0.05$). This supports a 1D association of the modules, since if the signal we detect were based on 3D contacts, one would expect a drop of BiFC positive cells in TSA treated cells. Second, we have now generated a cell line that stably expresses the BiAD 1 sensor and report additional results obtained with it in the new Fig. 7. In this cell line, the steady-state level of chromatin bound BiFC signal depends on the balance of fluorophore dissociation from chromatin and reconstitution of new fluorophores. At the resolution of our experiment, we detect constant levels of BiFC fluorescence as cells undergo mitotic division. We think that this strongly supports the 1D based reconstitutions mechanism as if the signal had been formed by 3D contacts, we should have observed strong changes in mitosis and/or cell toxicity.

Based on these arguments we are confident that, signals in our system strong originate from the adjacent docking of the anchor and detector modules in close 1D proximity and not through unspecific transient interactions or 3D contacts. We have included this line of argumentation in the discussion.

“It is also not proven that the sensor is associating on DNA rather than in a general nuclear region surrounding the target loci due to increased local concentration through weak interactions. Although the latter is still useful for detecting weak interactions, ChIP-qPCR at several loci if interest could be used as an orthogonal method to show convincingly that the two parts of the BiAD sensor are associating with the DNA at a level that can be detected by crosslinking. In any case, whether the sensor binds tightly to DNA or only weakly probes it should be discussed. In general, orthogonal "true positive" data (orthogonal genomic methods) need to be presented to ensure these experiments are working the way they are expected to work.”

Reply: Our control experiments with the fluorescently labelled DNA binding domains unequivocally demonstrate that the BiAD signal appear at the binding sites of the DNA binding domains. As described above, a ChIP approach cannot not useful in this regard. Our new data showing that the BiAD sensor 1 remains associated with mitotic chromosomes is an additional indicator of the tight binding of the sensor to chromatin. On the other hand, since the cells are able to proceed through cell division successfully, we think that the modules are

able to also transiently dissociate from chromatin. This aspect is now discussed in the discussion section.

“The evaluation of BiAD Sensor 2 could be more convincing if a pericentromere or centromere-associated protein was imaged by immunofluorescence at the same time as detecting the spotty BiFC signal – do they actually colocalize?”

Reply: We would like to draw the attention of the reviewer to the new Supplementary Fig. 10a. Here we have used the published and previously validated TALE-Venus (Ma et al., PNAS, 2013) construct to ensure that it co-localizes with the TALE-mRuby2 fusion generated in our study. Since the two constructs co-localized, we concluded that the TALE-mRuby2 can be used as a marker for pericentromeric spots, as shown by Ma et al. for the Venus fusion. Furthermore, we observed co-localization between TALE-mRuby2 and the BiFC signal as shown in the new Fig. 3b. This strongly indicates that the BiFC signals produced by BiAD sensor 2 occur at pericentromeric chromatin and are DNA sequence specific.

“Interactions between the split FPs could alter the association of genomic loci in living cells. The authors should discuss this potential source of artefacts in more detail. One way to discuss this would be using comparisons of BiFC signal and colocalization microscopy as shown in Fig. S3. Artefactual chromatin associations caused by the split FP would be subtle, so a quantitative analysis of the spot sizes and signal distribution for colocalization microscopy and BiFC could be helpful to evaluate whether this is a problem. It should be checked if the split FPs artefactually bridge chromosomal loci together, changing chromatin conformation in the cell. For single locus targeted sensors (such as the one on chromosome 9), 3C could be used to validate that 3-D chromosomal contacts don't appreciably change in the presence of the sensor.”

Reply: We agree that these are important questions and have discussed this point now. The proposed experiments are well thought and very interesting but they really go beyond the scope of this paper, demonstrating the applicability of the epigenetics BiFC sensor for the first time.

“How much time passed between transfection and imaging? There is only information on this for the 5-aza-dC experiment and the iMEF experiment in the methods (p. 19). For those, it looks like the time between transfection and imaging was 48 hours, which is longer than recommended (Kerppola, Nature Protocols, 2006). Why was such a long time point used, and could it lead to nonspecific signal?”

Reply: We would like to thank the reviewer for picking up this unintended omission. The time passed between transfection and fixation (48 h) has been clarified for all experiments now. The time point selection for imaging the cells was done based on the need to obtain clearly visible signals. The time needed for this depends on expression levels of the sensor parts and other peculiarities like nuclear transport and chromatin targeting that cannot be extrapolated from other experimental systems. As shown by the negative controls used as a reference (pocket mutants in the detector domains and cell lines with globally depleted DNA methylation or H3K9me3 marks), the signals we observed after 48 h are specific. This we have further validated by numerous control experiments throughout the paper.

In addition, we have selected Venus and the split site 210 due to its previously reported low tendency to non-specifically assemble (Ohashi et al., BioTechniques, 2012). This split site was not available at the time of the referred Kerppola paper but was shown to be superior in

terms of specificity and lack of spontaneous self-assembly (Ohashi et al., BioTechniques, 2012). In addition, the visibility of the BiFC signal is directly correlated with the abundance of the protein complexes under investigation. It is to be expected that BiFC assays addressing direct protein-protein contacts of abundant protein complexes would give rise earlier to fluorescent signals.

“Figure S4a, top two panels: The strong BiFC signal does not seem to colocalize with cells that have strong transfection control signal, so the figure panel does not support what is asserted in the caption. Many nuclei have BiFC signal where there is no visible signal in the transfection control channel. Can the authors elaborate on this, and maybe use thin white outlines to mark the cells that have strong transfection to clarify? Quantifying the total transfection control signal and total BiFC signal in each cell for a larger number of cells would also make this more interpretable.”

Reply: This point has been addressed above.

“Figure 3c: It is particularly important here to describe whether the contrast is the same in all panels.”

Reply: Contrast and settings are certainly always identical in connected images, as described above. This is now explicitly stated in the methods section and the figure legends.

“Figure S9c: Two cells per condition are not sufficient to evaluate if BiFC signal changed during treatment with TSA. Please include more images or a quantification as in Figure 3.”

Reply: A quantification of the TSA treatment has now been included in the new Supplementary Fig. 16d.

“Minor issues

p. 3, 2nd paragraph: “has not be achieved” should read “has not been achieved”

Reply: corrected.

“p. 3, 2nd paragraph: Non-affinity-based methods like bisulfite sequencing and methylation-sensitive PCR are also used to detect 5mC”

Reply: This has been rephrased.

“p. 3, last paragraph: several awkward phrasing and a typo “mintbodies””

Reply: Mintbody is not a typo but a term used for fluorescent modification-specific intracellular antibodies.

“p. 4, 2nd paragraph: What motivated the choice of the HP1-beta chromodomain?”

Reply: This was described in the results section ‘To detect H3K9me3, the chromo domain of HP1β (HP1CD) was selected, which retains the high H3K9me3-binding affinity of the full length protein (Bock et al. 2011) while lacking the SUV39H1-interacting chromoshadow domain (Yamamoto and Sonoda 2003).’

“p. 4, 3rd paragraph: acronym BiFC is not defined.”

Reply: Corrected.

“Fig. 1 caption, b,d: Fluorescence channel shown in red is not explained.”

Reply: Corrected.

“p. 6, 3rd paragraph: Typo in “to proof its 5mC specificity””

Reply: Corrected.

“p. 9, 2nd paragraph” “interested to explore” awkward phrasing”

Reply: This has been changed.

“p. 20: formatting error in reference 1: “Nature reviews. Genetics”. Same formatting issue in references 4 and 9.”

Reply: The formatting of references has been corrected.

“p. 23, ref. 44: The authors should cite the original work where the 7 nm distance is calculated or estimated, not a paper that cites other work for this number. From a brief look at the paper cited and the citations within that paper, it is not clear how the 7 nm distance is calculated.”

Reply: We agree with the reviewer. We have exchanged the reference to Hu, Huan, et al.

"Live visualization of genomic loci with BiFC-TALE." Scientific Reports 7 (2017): 40192, where this distance is estimated in more detail.

“Overall, standard error on the mean of 2 measurements should not be reported, just plot the results of the two measurements.”

Reply: We think this is an efficient and transparent way of reporting data. We have no objections to change the presentation, if this is requested by the editors.

Reply to Reviewer 2

“This manuscript presents an interesting approach to visualize the epigenetic status of specific DNA loci. This method relies on fusing a split fluorescent protein into two sensors, one detecting the DNA or Histone modification and the other detecting a specific DNA sequence. Upon binding of the sensors in close proximity the full fluorescent protein is reconstituted and signal can be detected. The authors present two “epigenetic” sensors: one detecting DNA methylation and one detecting the H3K9me3 modification. We recognize the innovative component of this system and its potential for live imaging and tracking of epigenetic modification. In addition, we appreciate the fact that this method is able to detect changes in the enrichment of these epigenetic modifications.”

Reply: Thank you very much for these positive statements.

“However, as presented the method still falls short of its main goals. The advantage of this system compared to currently existing methods (such as PLA) is the ability to track changes that are occurring in living cells. However, the authors do not show any videos of their experiments. How long is it possible to track these cells for? Can the cells be followed through cell division? Is the signal lost during mitosis? The authors should show videos and representative images of cells over one to two days. This should be done for both DNA methylation and H3K9me3.”

Reply: We thank the reviewer for this piece of constructive criticism which motivated us to include several additional experiments addressing this point (new Fig. 7).

First, we would like to mention that the aim of this paper was to provide the scientific community with a novel set of tools that allow the site-specific investigation of candidate target marks. For this reason, we have mainly resorted to transient transfection experiments coupled with fixation as a rapid screening and tool development approach. We have now included the new Supplementary Fig. 3 to document that the BiFC signal is also visible and specific in live cells.

Second, we have now generated a cell line that stably express the BiAD sensor 1. With this we were able to track DNA methylation levels during cell division (mitosis) and 5-aza-dC treatment. Both experiments were successful and clearly document that the BiAD tools can be used for the dynamic imaging of epigenetic marks in live cells, as requested by the reviewer.

“As the authors point out in their discussion this work is still limited to visualization of large repetitive regions. As such, almost all figures shown here could have been done with the sensors that detect the epigenetic modifications. The only two figures that have some specificity at the level of DNA loci are 2D and 3F where the authors use dCAS9 to label pericentromeric repeats on chromosome 19. The authors should show that their system is able to detect a single locus. This could be done by targeting single loci that contain several repeats that could be bound by a single gRNA (like the MUC or IGH genes) but also by targeting several sgRNAs to one location. Unless the method really is able to allow visualization of specific single loci (instead of repeats) we do not believe it will be useful for the community in a way that grants publication in Nature Communications.”

Reply: While we agree that these are interesting directions of development we think that these experiments are well beyond the scope of the current manuscript as they will require careful design, validation and optimization of all parts of the system to ensure functionality. Furthermore, we would like to highlight that while we address sites with different degrees of

repetitiveness, all of our sensors have specificity at the level of DNA loci as shown by numerous controls.

In addition, as clearly documented in new Supplementary Fig. 13 by a side-by-side comparison between co-localisation microscopy and BiAD signals, not only BiAD sensor 3 but also BiAD sensor 2, constructed with the TALE module, produces a visual output that cannot be obtained with a sensor that only detects DNA methylation independent of DNA sequence. Taking this into account actually not 2 but 10 main figure panels of this paper would have been impossible to obtain with domains that detect only epigenetic modifications. While we agree with the reviewer that the MUC and IGH genes would be interesting targets to visualize, as already done in other microscopy-based studies, these sequences are by themselves repetitive and therefore conceptually not very different than the targets we have already looked at.

Finally, the aim of our paper was to demonstrate that BiFC-based sensors are a viable solution to simultaneously access the DNA sequence and epigenetic mark information in living cells. So far, this is the only methodology meeting this experimental demand and therefore we are convinced that it is useful to the scientific community. We have no doubt that the BiFC reconstitution event will also take place at the level of single copy loci and that this can be readout with microscopes that are more sensitive than the standard laser scanning microscope available at our facility. As indicated in the discussion section of the paper, the feasibility of this approach was already demonstrated by Chen, et al. *ACS nano*, 2016, who combine fluorescence complementation with PALM to detect protein complexes with single molecule sensitivity.

“Could the authors demonstrate that the system works with active histone marks like H3K27ac or H3K36me3 – use of this system for tracking an actively transcribing gene is important.”

Reply: We agree, but we like to mention, that we have pioneered the development of these systems for two different chromatin marks. Certainly others will follow, but this will require careful design, validation and optimization of all parts of the system to ensure functionality and is well beyond the scope of the current manuscript.

“Finally, the intensity of the signals shown in figure 4a for the H3K9me3 modification is a bit worrisome. The signal to noise ratio seems very low and cells seem to have been exposed for quite some time. If that is the case does it become toxic?”

Reply: The reviewer is right that the sensor described in Fig. 5 is less bright than the 5mC sensors, which made necessary a corresponding adjustment of the imaging parameters. This reduced brightness of the H3K9me3 sensor is now mentioned in the methods part of the manuscript. However, we would like to mention that despite its reduced brightness, the sensitivity to H3K9me3 modifications (in terms of signal to noise ratio) was comparable to that of the 5mC sensors towards 5mC, which we believe is an important point for the application of this tool (new Fig. 2d vs 6c).

We did not find sign of toxicity in the experiments with sensors 1, 3 and 4. However, we observed that the TALE domain is critical and must not be expressed too high, but this was equally true for the single TALE and the TALE being part of the BiFC system. The newly added live cell imaging data clearly illustrate that the sensors are not toxic to cells when used under the conditions of our study.

“Other points:

All figures where DAPI is used to validate a finding should be shown with the merged channels to allow proper evaluation. In general all figures should have a merged version.”

Reply: In our impression adding a merge will reduce the sizes of the relevant panels, making it more difficult for readers to see the effects. We have however added, where relevant for data interpretation, additional merges of the figures. Merges are now shown in new Fig. 2b, and new Supplementary Figs. 1, 2, 4, 5, 6, 7, 8, 9, 10a, 11a , b, 13a, 14, 20, 21, 22, 23a and 25a

“All the figure callouts are in totally random order – this is extremely annoying. Figures should be reorganized to follow the text.”

Reply: We thank the reviewer for this suggestion; the figures have now been reorganized.

REVIEWERS' COMMENTS:

Reviewer #1 (Remarks to the Author):

I am satisfied with the authors responses.

Reviewer #2 (Remarks to the Author):

The revised manuscript by Lungu and colleagues is significantly improved from its first version. I appreciate that the figure order is now easier to read and am also pleased by the addition of data showing how cells can be tracked throughout cell cycle. The addition of figure 7 where cells are tracked through 5-aza treatment and recovery is a significant improvement.

My request to add a repeat region different from satellites or centromeres was to show that the system is able to detect smaller loci. Although the manuscript is now improved from the first version I am disappointed that the authors did not attempt this very straight forward experiment. I still believe that adding this feature and an additional activating histone mark would broaden the scope for the use of this technique.

Minor point: Figure 1 should include the information shown in table 1 so that the reader can immediately understand the different combinations of DNA and protein sensors used in this manuscript and also know what each BiAD sensor is able to detect.

Reply to the comment of reviewer 2

The revised manuscript by Lungu and colleagues is significantly improved from its first version. I appreciate that the figure order is now easier to read and am also pleased by the addition of data showing how cells can be tracked throughout cell cycle. The addition of figure 7 where cells are tracked through 5-aza treatment and recovery is a significant improvement.

Reply: Thank you for this positive assessment.

My request to add a repeat region different from satellites or centromeres was to show that the system is able to detect smaller loci. Although the manuscript is now improved from the first version I am disappointed that the authors did not attempt this very straight forward experiment. I still believe that adding this feature and an additional activating histone mark would broaden the scope for the use of this technique.

Reply: We agree, but as stated in our previous response, setting up an additional new sensor would constitute a new research project. It would require numerous of controls, similarly as shown here for the 5mC and H3K9me3 sensors and this is far beyond the scope of the current manuscript.

Minor point: Figure 1 should include the information shown in table 1 so that the reader can immediately understand the different combinations of DNA and protein sensors used in this manuscript and also know what each BiAD sensor is able to detect.

Reply: The table has been incorporated into Fig. 1 so that the description and schematic drawing are presented together at the beginning of the paper.